# Sparse Bayesian Generative Modeling for Compressive Sensing

**Benedikt Böck, Sadaf Syed, Wolfgang Utschick**
TUM School of Computation, Information and Technology
Technical University of Munich
{benedikt.boeck,sadaf.syed,utschick}@tum.de

## Abstract

This work addresses the fundamental linear inverse problem in compressive sensing (CS) by introducing a new type of regularizing generative prior. Our proposed method utilizes ideas from classical dictionary-based CS and, in particular, sparse Bayesian learning (SBL), to integrate a strong regularization towards sparse solutions. At the same time, by leveraging the notion of conditional Gaussianity, it also incorporates the adaptability from generative models to training data. However, unlike most state-of-the-art generative models, it is able to learn from a few compressed and noisy data samples and requires no optimization algorithm for solving the inverse problem. Additionally, similar to Dirichlet prior networks, our model parameterizes a conjugate prior enabling its application for uncertainty quantification. We support our approach theoretically through the concept of variational inference and validate it empirically using different types of compressible signals.

## 1 Introduction

Research in CS has shown that it is possible to reduce the number of measurements far below the one determined by the Nyquist sampling theorem while still being able to extract the information-carrying signal from the acquired observations. The fundamental problem in CS is an ill-posed linear inverse problem, i.e., the goal is to recover the signal $\boldsymbol{x}^* \in \mathbb{R}^N$ of interest from an under-determined set of measurements $\boldsymbol{y} \in \mathbb{R}^M$ with $M \ll N$, related by

$$\boldsymbol{y} = \boldsymbol{A}\boldsymbol{x}^* + \boldsymbol{n}, \tag{1}$$

where $\boldsymbol{x}^*$ is compressed by the measurement matrix $\boldsymbol{A}$ and potentially corrupted by additive noise $\boldsymbol{n} \sim \mathcal{N}(\boldsymbol{0}, \sigma_n^2 \mathbf{I})$. Since the under-determined observation $\boldsymbol{y}$ does not carry enough information alone to faithfully reconstruct $\boldsymbol{x}^*$, additional prior (or model) knowledge about $\boldsymbol{x}^*$ is required to make the inverse problem "well-posed". Classical CS algorithms such as Lasso regression or orthogonal matching pursuit (OMP) address this problem by incorporating the model knowledge that $\boldsymbol{x}^*$ is sparse or compressible with respect to some dictionary [1]. Nowadays, modern deep learning (DL)-based approaches such as unfolding algorithms, generative model-based CS and un-trained neural networks (NNs) expand the possibilities by learning prior knowledge from a training set or designing the network architecture to be biased towards a certain class of signals [2–7]. These approaches typically require a (potentially large) training set of ground-truth data samples, or their architecture is specifically biased towards natural images. However, in many applications, ground-truth training data might not be easily accessible. In, e.g., electron microscopy, the amount of electron dose has to be restricted to not induce damage on the probe resulting in low-contrast noisy data samples [8]. The sensors in wearable electrocardiography (ECG) monitoring devices generally provide noisy signals with artifacts [9], which limits the ability to learn from patient-specific data in real-world settings. Another example is the wireless 5G communication standard, where mobile users receive compressed and noisy so-called channel observations on a frequent basis (cf. [10]) while acquiring ground-truth

38th Conference on Neural Information Processing Systems (NeurIPS 2024).

channel information requires costly measurement campaigns. Thus, in many applications, it is either impossible or prohibitively expensive to collect lots of ground-truth training data while corrupted data is readily available. This highlights the necessity for methods that can learn from corrupted data samples.

In this paper, we propose a new learnable prior for solving the inverse problem (1), which can learn from only a few compressed and noisy data samples and, thus, requires no ground-truth information in its training phase. Additionally, it applies to any type of signal, which is compressible with respect to some dictionary. Our approach shares similarities with generative model-based CS [4, 11]. There, a generative model is first trained to capture the exact underlying prior distribution $p(\boldsymbol{x})$ of the signal $\boldsymbol{x}^*$ of interest. It then serves as probabilistic prior to regularize the inverse problem (1). Similarly, classical CS algorithms like Lasso regression also impose a probabilistic prior on the sparse representation of $\boldsymbol{x}^*$ with respect to some dictionary [12]. In contrast to modern generative models, however, these priors do not have any generation capabilities but solely bias the inverse problem (1) towards sparse solutions. This observation indicates that a probabilistic prior does not necessarily need to capture the exact prior distribution $p(\boldsymbol{x})$ to effectively regularize (1). The proposed model in this work builds upon this insight and forms a trainable but simultaneously sparsity-inducing prior. For that, we aim to combine the adaptability of generative models to training data with the property of many types of signals $\boldsymbol{x}^*$ to be compressible with respect to some dictionary. As a result, our model learns statistical information in the signal's sparse/compressible domain. Examples of signals with a specific statistical structure in their compressible representation include piecewise smooth functions and natural images, whose wavelet coefficients approximately build a connected sub-tree or wireless channels, which are burst-sparse in their angular domain [13, 14].

**Related Work.** Early work on CS, which considers statistical structure in the wavelet domain of images, is given in [15, 16]. Based on the theoretical foundation in [17] and the concept of SBL [18, 19], these papers introduce a hierarchical Bayesian model, which is used to apply variational inference and Markov chain Monte Carlo (MCMC) posterior sampling to solve (1). Training Gaussian mixture models (GMMs), i.e., classical generative models, from compressed image patches of one or a few images has been analyzed in [20–23]. In this line of research, however, the GMM is fit directly in the pixel domain. Compressive dictionary learning represents a different line of research aiming to learn the dictionary from solely compressed data [24–27], and strongly bases on the dictionary learning method K-SVD [28]. More recently, variational autoencoders (VAEs) and generative adversarial networks (GANs) have been studied in the context of CS [4, 11]. There, the VAE/GAN is used to solve the inverse problem (1) by constraining the signal $\boldsymbol{x}^*$ of interest to lie in the range of the generative model instead of being sparse with respect to some dictionary. In this context, AmbientGAN as well as CSGAN are extensions that loosen up the training set requirements and can learn from corrupted data [29, 30]. Another related topic is the ability of some generative models, i.e., diffusion models, VAEs and GMMs, to provide an approximation of the conditional mean estimator (CME) [31–33]. In the case of the latter two, the CME is represented as a tractable convex combination of linear minimum mean squared error (MSE) estimators by exploiting their conditional Gaussianity on a latent space that determines these estimators' means and covariances.

**Our main contributions are as follows:**

- We introduce a new type of sparsity-inducing generative prior for the inverse problem (1), which differs from classical CS algorithms due to its ability to learn from data. On the other hand, it also differs from other modern NN-based approaches due to its ability to promote sparsity in the signal's compressible domain. Moreover, it can learn from a few corrupted data samples and, thus, requires no ground-truth information in its training phase.

- We theoretically underpin our approach by proving that its training maximizes a variational lower bound of a sparsity-inducing log-evidence.

- Building on the notion of conditional Gaussianity, we introduce two specific implementations of the proposed type of prior based on VAEs and GMMs, which do not require an optimization algorithm in their inference phase and come with computational benefits.

- By exploiting the shared property with Dirichlet prior networks to parameterize a conjugate prior, we demonstrate how our approach can be applied for uncertainty quantification.

- We validate the performance on datasets containing different types of compressible signals.[1]

---

[1]Source code is available at `https://github.com/beneboeck/sparse-bayesian-gen-mod`.

## 2 Background and Method

**Notation.** The operations $\boldsymbol{b}^{-1}$, $\sqrt{\boldsymbol{b}}$ and $|\boldsymbol{b}|^2$ denote the respective element-wise operation for the vector $\boldsymbol{b}$. Moreover, $\mathrm{diag}(\boldsymbol{B})$ represents the vectorized diagonal of the matrix $\boldsymbol{B}$, while $\mathrm{diag}(\boldsymbol{b})$ denotes the diagonal matrix with the vector $\boldsymbol{b}$ on its diagonal.

### 2.1 Problem Formulation

We consider the typical CS setup, in which we measure $N$-dimensional ground-truth samples $\boldsymbol{x}_i$ by a known measurement matrix $\boldsymbol{A} \in \mathbb{R}^{M \times N}$ (or $\boldsymbol{A}_i \in \mathbb{R}^{M \times N}$) with $M \ll N$, where each observation is potentially corrupted by noise $\boldsymbol{n}_i$ drawn from additive white Gaussian noise (AWGN) $\boldsymbol{n} \sim \mathcal{N}(\boldsymbol{0}, \sigma_n^2 \mathbf{I})$. We assume all $\boldsymbol{x}_i$ to be compressible with respect to a known dictionary matrix $\boldsymbol{D} \in \mathbb{R}^{N \times S}$, i.e., $\boldsymbol{y}_i = \boldsymbol{AD}\boldsymbol{s}_i + \boldsymbol{n}_i$ with $\boldsymbol{x}_i = \boldsymbol{D}\boldsymbol{s}_i$ and $\boldsymbol{s}_i \in \mathbb{R}^S$ being approximately sparse. All $\boldsymbol{s}_i$ are assumed to be independent and identically distributed (i.i.d.), i.e., $\boldsymbol{s}_i \sim p(\boldsymbol{s})$, where $p(\boldsymbol{s})$ is unknown with $\boldsymbol{s}$ exhibiting non-trivially dependent entries. Typical signals which fit into this category are, e.g., natural images, piecewise smooth functions, and wireless channels (cf. Section 1). Our approach in this work allows to either solely have access to corrupted training observations $\mathcal{Y} = \{\boldsymbol{y}_i\}_{i=1}^{N_t}$ or training tuples with ground-truth information $\mathcal{G} = \{(\boldsymbol{s}_i, \boldsymbol{y}_i)\}_{i=1}^{N_t}$ or $\mathcal{W} = \{(\boldsymbol{x}_i, \boldsymbol{y}_i)\}_{i=1}^{N_t}$. We first train the proposed model using $\mathcal{Y}$ (or, alternatively, $\mathcal{G}$ or $\mathcal{W}$) to serve as an effective prior for (1). Our goal is then to estimate a ground-truth signal $\boldsymbol{x}^*$ of a newly observed $\boldsymbol{y}$. We also define $\mathcal{X} = \{\boldsymbol{x}_i\}_{i=1}^{N_t}$ and $\mathcal{S} = \{\boldsymbol{s}_i\}_{i=1}^{N_t}$.

### 2.2 Sparse Bayesian Learning for Compressive Sensing

In SBL for CS, the idea for solving the inverse problem (1) is to assign a parameterized prior to $\boldsymbol{s}^*$, i.e., the compressible representation of the signal $\boldsymbol{x}^*$ of interest with $\boldsymbol{x}^* = \boldsymbol{D}\boldsymbol{s}^*$ [19]. It assumes

$$\boldsymbol{y}|\boldsymbol{s}^* \sim p(\boldsymbol{y}|\boldsymbol{s}^*) = \mathcal{N}(\boldsymbol{y}; \boldsymbol{AD}\boldsymbol{s}^*, \sigma^2 \mathbf{I}), \quad \boldsymbol{s}^* \sim p_{\boldsymbol{\gamma}}(\boldsymbol{s}) = \mathcal{N}(\boldsymbol{s}; \boldsymbol{0}, \mathrm{diag}(\boldsymbol{\gamma})). \tag{2}$$

Given a single observation $\boldsymbol{y}$, the parameters $\boldsymbol{\gamma}$ (and sometimes $\sigma^2$) are estimated by an expectation-maximization (EM) algorithm maximizing the corresponding log-evidence $\log p_{\boldsymbol{\gamma}}(\boldsymbol{y})$ implicitly defined by (2). After that, it is utilized that $p_{\boldsymbol{\gamma}}(\boldsymbol{s})$ forms a conjugate prior of $p(\boldsymbol{y}|\boldsymbol{s}^*)$ resulting in a closed-form posterior $p_{\boldsymbol{\gamma}}(\boldsymbol{x}|\boldsymbol{y})$ providing all necessary information to estimate $\boldsymbol{x}^*$. In [19], a variational interpretation of SBL is given, addressing why this approach yields sparse results, even though $p_{\boldsymbol{\gamma}}(\boldsymbol{s})$ does not inherently promote sparsity. It is shown that there exists a $C > 0$ such that for all $\boldsymbol{\gamma} > \boldsymbol{0}$ and $\boldsymbol{s}$

$$p_{\boldsymbol{\gamma}}(\boldsymbol{s}) \leq t(\boldsymbol{s}) = C \cdot \prod_{i=1}^{S} \frac{1}{|s_i|}. \tag{3}$$

The function $t(\boldsymbol{s})$, however, is a well-known improper but sparsity-inducing prior, used as a non-informative prior for scale parameters [34]. Let another statistical model be given by $p(\boldsymbol{y}|\boldsymbol{s}^*)$ (cf. (2)) and prior $t(\boldsymbol{s})$ instead of $p_{\boldsymbol{\gamma}}(\boldsymbol{s})$ with implicitly defined log-evidence $\log \pi^{(s)}(\boldsymbol{y})$. Due to $t(\boldsymbol{s})$ being improper, the log-evidence $\log \pi^{(s)}(\boldsymbol{y})$ forms a not normalized but valid log-likelihood. Moreover, this model is sparsity-inducing due to the sparsity-promoting characteristics of $t(\boldsymbol{s})$. Based on (3), it holds that $\log \pi^{(s)}(\boldsymbol{y}) \geq \log p_{\boldsymbol{\gamma}}(\boldsymbol{y})$ and, thus, $\log p_{\boldsymbol{\gamma}}(\boldsymbol{y})$ can be embedded in variational inference, forming a tractable variational lower bound of an intractable log-evidence $\log \pi^{(s)}(\boldsymbol{y})$ of actual interest. Moreover, applying the EM algorithm to (2) is "evidence maximization over the space of variational approximations to a model (i.e., $p(\boldsymbol{y}|\boldsymbol{s}^*)$ and $t(\boldsymbol{s})$), with a sparse, regularizing prior" [19].

### 2.3 Gaussian Mixture Models and Variational Autoencoders

GMMs and VAEs are generative models aiming to learn an unknown distribution $p(\boldsymbol{x})$ from a training set $\mathcal{X}$. Both models represent $p(\boldsymbol{x})$ as the marginalization with conditionally Gaussian $p(\boldsymbol{x}|\cdot)$ and an additional latent variable $k$ (or $\boldsymbol{z}$) [35–37], i.e.,

$$p^{(\mathrm{GMM})}(\boldsymbol{x}) = \sum_{k=1}^{K} p(k)p(\boldsymbol{x}|k) = \sum_{k=1}^{K} \rho_k \mathcal{N}(\boldsymbol{x}; \boldsymbol{\mu}_k, \boldsymbol{C}_k), \tag{4}$$

$$p^{(\mathrm{VAE})}(\boldsymbol{x}) = \int p(\boldsymbol{z})p_{\boldsymbol{\theta}}(\boldsymbol{x}|\boldsymbol{z})\mathrm{d}\boldsymbol{z} = \int \mathcal{N}(\boldsymbol{z}; \boldsymbol{0}, \mathbf{I})\mathcal{N}(\boldsymbol{x}; \boldsymbol{\mu}_{\boldsymbol{\theta}}(\boldsymbol{z}), \boldsymbol{C}_{\boldsymbol{\theta}}(\boldsymbol{z}))\mathrm{d}\boldsymbol{z}. \tag{5}$$

The tunable parameters ($\{\rho_k, \boldsymbol{\mu}_k, \boldsymbol{C}_k\}_{k=1}^{K}$ for GMMs and $\boldsymbol{\theta}$ for VAEs) are learned by optimizing the model's log-evidence $\sum_i \log p^{(f)}(\boldsymbol{x}_i)$ ($f \in \{\text{GMM}, \text{VAE}\}$) over $\mathcal{X}$. In the case of GMMs, this is typically done by an EM algorithm alternating between the so-called e- and m-step. The e-step determines the closed-form posteriors $p_t(k|\boldsymbol{x}_i)$ (for all $i$) via the Bayes rule using the model's parameters in the $t$th iteration. The m-step then updates the model's parameters by maximizing $\sum_i \mathbb{E}_{p_t(k|\boldsymbol{x}_i)}[\log p(\boldsymbol{x}_i, k)]$. In case of VAEs, however, the posterior $p_{\boldsymbol{\theta}}(\boldsymbol{z}|\boldsymbol{x})$ (and $\log p^{(\text{VAE})}(\boldsymbol{x})$) are intractable and the EM algorithm cannot be applied. Therefore, a tractable distribution $q_{\boldsymbol{\phi}}(\boldsymbol{z}|\boldsymbol{x}) = \mathcal{N}(\boldsymbol{z}; \boldsymbol{\mu}_{\boldsymbol{\phi}}(\boldsymbol{x}), \text{diag}(\boldsymbol{\sigma}_{\boldsymbol{\phi}}^2(\boldsymbol{x}))$ with variational parameters $\boldsymbol{\phi}$ is introduced, which approximates $p_{\boldsymbol{\theta}}(\boldsymbol{z}|\boldsymbol{x})$, and $\boldsymbol{\mu}_{\boldsymbol{\phi}}(\boldsymbol{x})$ and $\boldsymbol{\sigma}_{\boldsymbol{\phi}}^2(\boldsymbol{x})$ are generated by a NN encoder. Equivalently, $\boldsymbol{\mu}_{\boldsymbol{\theta}}(\boldsymbol{z})$ and $\boldsymbol{C}_{\boldsymbol{\theta}}(\boldsymbol{z})$ are generally realized by a NN decoder. The objective to be maximized is the evidence lower bound (ELBO) $L(\boldsymbol{\theta}, \boldsymbol{\phi})$ serving as a tractable lower bound for the intractable log-evidence $\sum_i \log p_{\boldsymbol{\theta}}(\boldsymbol{x}_i)$, i.e.,

$$\sum_{\boldsymbol{x}_i \in \mathcal{X}} \log p_{\boldsymbol{\theta}}(\boldsymbol{x}_i) \geq L(\boldsymbol{\theta}, \boldsymbol{\phi}) = \sum_{\boldsymbol{x}_i \in \mathcal{X}} \left( \log p_{\boldsymbol{\theta}}(\boldsymbol{x}_i) - \text{D}_{\text{KL}}(q_{\boldsymbol{\phi}}(\boldsymbol{z}|\boldsymbol{x}_i) || p_{\boldsymbol{\theta}}(\boldsymbol{z}|\boldsymbol{x}_i)) \right) \tag{6}$$

with $\text{D}_{\text{KL}}(\cdot)$ being the Kullback-Leibler (KL) divergence. Generally, the GMM's and VAE's training critically depends on iteratively characterizing and updating their posteriors $p_t(k|\boldsymbol{x}_i)$ and $q_{\boldsymbol{\phi}}(\boldsymbol{z}|\boldsymbol{x}_i)$.

## 2.4 Proposed Method

The goal of this section is to derive a class of generative models, for which we can guarantee that, on the one hand, it is sparsity-inducing, but on the other hand, it is trainable and can learn from solely compressed and noisy observations $\mathcal{Y}$ resulting in an effective probabilistic prior for (1).

To incorporate the bias towards sparsity, we start with SBL discussed in Section 2.2 and combine it with the VAE's and GMM's main principle for their adaptability to complicated distributions, i.e., introducing a latent variable $\boldsymbol{z}$ (or $k$) on which we condition with a parameterized Gaussian (cf. (4) and (5)).[2] More specifically, we exploit the Gaussianity of $p_{\boldsymbol{\gamma}}(\boldsymbol{s})$ in (3) and modify it to a parameterized Gaussian conditioned on some latent variable $\boldsymbol{z}$ with arbitrarily parameterized $p_{\boldsymbol{\delta}}(\boldsymbol{z})$ while explicitly keeping its mean zero and its covariance matrix diagonal, i.e., $p_{\boldsymbol{\theta}}(\boldsymbol{s}|\boldsymbol{z}) = \mathcal{N}(\boldsymbol{s}; \boldsymbol{0}, \text{diag}(\boldsymbol{\gamma}_{\boldsymbol{\theta}}(\boldsymbol{z})))$. The resulting set of statistical models referred to as sparse Bayesian generative models, is given by

$$\boldsymbol{y}|\boldsymbol{s} \sim p(\boldsymbol{y}|\boldsymbol{s}) = \mathcal{N}(\boldsymbol{y}; \boldsymbol{A}\boldsymbol{D}\boldsymbol{s}, \sigma^2 \mathbf{I}), \;\; \boldsymbol{s}|\boldsymbol{z} \sim p_{\boldsymbol{\theta}}(\boldsymbol{s}|\boldsymbol{z}) = \mathcal{N}(\boldsymbol{s}; \boldsymbol{0}, \text{diag}(\boldsymbol{\gamma}_{\boldsymbol{\theta}}(\boldsymbol{z}))), \;\; \boldsymbol{z} \sim p_{\boldsymbol{\delta}}(\boldsymbol{z}). \tag{7}$$

**Training principle.** Our proposed training scheme is independent of specific realizations for $\boldsymbol{\gamma}_{\boldsymbol{\theta}}(\boldsymbol{z})$ and $p_{\boldsymbol{\delta}}(\boldsymbol{z})$, which is why they are kept general in this section. Specific parameterizations are discussed in Section 3.2 and 3.3. The training goal is to maximize the model's log-evidence $\sum_i \log p_{\boldsymbol{\delta}, \boldsymbol{\theta}}(\boldsymbol{y}_i)$ implicitly defined by (7) over a training set of compressed and potentially noisy observations $\mathcal{Y}$. For that, we rely on the main training principles for statistical models including latent variables, i.e., the EM algorithm and variational inference (cf. Section 2.3). A key requirement of these principles is the ability to track and update the model's posterior (i.e., $p_{\boldsymbol{\theta}, \boldsymbol{\delta}}(\boldsymbol{s}, \boldsymbol{z}|\boldsymbol{y})$ for (7)) or approximations of it over the training iterations (cf. Section 2.3). In classical SBL, the prior distribution $p_{\boldsymbol{\gamma}}(\boldsymbol{s})$ forms a conjugate prior of $p(\boldsymbol{y}|\boldsymbol{s})$ (cf. (2)) and, thus, the posterior $p_{\boldsymbol{\gamma}}(\boldsymbol{s}|\boldsymbol{y})$ is tractable in closed form. We utilize this property by observing that conditioned on some $\boldsymbol{z}$, (7) coincides with the classical SBL model. Consequently, $\boldsymbol{s}|\boldsymbol{z}$ is a conditioned conjugate prior of $\boldsymbol{y}|\boldsymbol{s}$ in (7) with tractable Gaussian posterior $p_{\boldsymbol{\theta}}(\boldsymbol{s}|\boldsymbol{z}, \boldsymbol{y})$, whose closed-form covariance matrix and mean

$$\boldsymbol{C}_{\boldsymbol{\theta}}^{\boldsymbol{s}|\boldsymbol{y}, \boldsymbol{z}}(\boldsymbol{z}) = \left( \frac{1}{\sigma^2} \boldsymbol{D}^{\text{T}} \boldsymbol{A}^{\text{T}} \boldsymbol{A} \boldsymbol{D} + \text{diag}(\boldsymbol{\gamma}_{\boldsymbol{\theta}}^{-1}(\boldsymbol{z})) \right)^{-1}, \; \boldsymbol{\mu}_{\boldsymbol{\theta}}^{\boldsymbol{s}|\boldsymbol{y}, \boldsymbol{z}}(\boldsymbol{z}) = \frac{1}{\sigma^2} \boldsymbol{C}_{\boldsymbol{\theta}}^{\boldsymbol{s}|\boldsymbol{y}, \boldsymbol{z}}(\boldsymbol{z}) \boldsymbol{D}^{\text{T}} \boldsymbol{A}^{\text{T}} \boldsymbol{y} \tag{8}$$

equal those in SBL [19]. The equivalent formulas for $\sigma^2 = 0$ are given in Appendix C. By considering the decomposition $p_{\boldsymbol{\theta}, \boldsymbol{\delta}}(\boldsymbol{s}, \boldsymbol{z}|\boldsymbol{y}) = p_{\boldsymbol{\theta}}(\boldsymbol{s}|\boldsymbol{z}, \boldsymbol{y}) p_{\boldsymbol{\theta}, \boldsymbol{\delta}}(\boldsymbol{z}|\boldsymbol{y})$, we conclude that our proposed set of statistical models in (7) exhibits posteriors, which are partially tractable in closed form independent of any specifics of $\boldsymbol{\gamma}_{\boldsymbol{\theta}}(\boldsymbol{z})$ and $p_{\boldsymbol{\delta}}(\boldsymbol{z})$. The remaining $p_{\boldsymbol{\theta}, \boldsymbol{\delta}}(\boldsymbol{z}|\boldsymbol{y})$ resembles the standard GMM's and VAE's posterior, and we can either apply the EM algorithm or variational inference to maximize the corresponding log-evidence. Both will be discussed in Section 3.2 and 3.3, respectively.

---

[2]For the sake of readability, we exclusively use $\boldsymbol{z}$ throughout the remainder of this section to denote the continuous as well as the discrete and finite random variable, on which we condition.

**Estimation scheme.** The goal after the training is to use the learned statistical model in (7) for regularizing the inverse problem (1) and estimating $\boldsymbol{x}^*$ from a newly observed $\boldsymbol{y}$. Generally, the CME $\mathbb{E}[\boldsymbol{x}|\boldsymbol{y}]$ is a desirable estimator due to its property of minimizing the MSE. We utilize insights from [33, 23], which derive approximate closed forms of the CME using VAEs and GMMs. More precisely, by using the law of total expectation, we approximate the CME by

$$\mathbb{E}[\boldsymbol{x}|\boldsymbol{y}] = \mathbb{E}\left[\mathbb{E}\left[\boldsymbol{x}|\boldsymbol{y},\boldsymbol{z}\right]|\boldsymbol{y}\right] = \boldsymbol{D}\,\mathbb{E}\left[\mathbb{E}\left[\boldsymbol{s}|\boldsymbol{y},\boldsymbol{z}\right]|\boldsymbol{y}\right] \approx \boldsymbol{D}\,\mathbb{E}_{p_{\boldsymbol{\theta},\boldsymbol{\delta}}(\boldsymbol{z}|\boldsymbol{y})}\left[\mathbb{E}_{p_{\boldsymbol{\theta}}(\boldsymbol{s}|\boldsymbol{z},\boldsymbol{y})}\left[\boldsymbol{s}|\boldsymbol{y},\boldsymbol{z}\right]|\boldsymbol{y}\right]. \quad (9)$$

The inner expectation is given by the learned $\boldsymbol{\mu}_{\boldsymbol{\theta}}^{\boldsymbol{s}|\boldsymbol{y},\boldsymbol{z}}(\boldsymbol{z})$ in (8), while, depending on its specific characteristics, $p_{\boldsymbol{\theta},\boldsymbol{\delta}}(\boldsymbol{z}|\boldsymbol{y})$ also exhibits a closed form or a variational approximation has been learned during training (cf. Section 3.2), with which we approximate the outer expectation by a Monte-Carlo estimation. Based on the perception-distortion trade-off, the CME is not optimal from a perceptual point [38], and deviating estimators from the CME approximation might be beneficial. More detailed descriptions of the CME approximation as well as further estimators based on the specific implementations in Section 3.2 and 3.3 are given in Appendix G.

# 3 Theoretical Analysis and Specific Parameterizations

## 3.1 Sparsity Guarantees

In Section 2.4, we motivate constraining $p_{\boldsymbol{\theta}}(\boldsymbol{s}|\boldsymbol{z})$ to be a zero mean conditional Gaussian with diagonal covariance matrix by SBL and its sparsity-inducing property. In the following, we rigorously show that constraining $p_{\boldsymbol{\theta}}(\boldsymbol{s}|\boldsymbol{z})$ in this manner is sufficient to maintain the sparsity-inducing property for any statistical model following (7) despite the additionally included latent variable $\boldsymbol{z}$. More precisely, we show that any statistical model in (7) with some specified parameterized $p_{\boldsymbol{\delta}}(\boldsymbol{z})$ and $\boldsymbol{\gamma}_{\boldsymbol{\theta}}(\boldsymbol{z})$ exhibits a log-evidence that is interpretable as a variational lower bound of a sparsity-inducing log-evidence. For that, we establish the following theorem.

**Theorem 3.1.** *Let $p_{\boldsymbol{\theta}}(\boldsymbol{s}|\boldsymbol{z}) = \mathcal{N}(\boldsymbol{s};\boldsymbol{0},diag(\boldsymbol{\gamma}_{\boldsymbol{\theta}}(\boldsymbol{z})))$ (i.e., it is defined according to (7)), let $\boldsymbol{z}$ be either continuous or discrete and finite, and let $\boldsymbol{\gamma}_{\boldsymbol{\theta}}(\boldsymbol{z}) > \boldsymbol{0}$. Then, there exists a constant $C > 0$ such that for all $\boldsymbol{s},\boldsymbol{\theta},\boldsymbol{\delta}$ with any arbitrary distribution $p_{\boldsymbol{\delta}}(\boldsymbol{z})$*

$$p_{\boldsymbol{\theta},\boldsymbol{\delta}}(\boldsymbol{s}) = \int p_{\boldsymbol{\delta}}(\boldsymbol{z})\mathcal{N}(\boldsymbol{s};\boldsymbol{0},diag(\boldsymbol{\gamma}_{\boldsymbol{\theta}}(\boldsymbol{z})))\mathrm{d}\boldsymbol{z} \leq t(\boldsymbol{s}) = C \cdot \prod_{i=1}^{N} \frac{1}{|s_i|}. \quad (10)$$

*The integral corresponds to a summation for discrete and finite $p_{\boldsymbol{\delta}}(\boldsymbol{z})$.*

The proof is given in Appendix A. Based on Theorem 3.1, it holds that

$$\left(\log \pi^{(s)}(\boldsymbol{y}) \geq \log p_{\boldsymbol{\theta},\boldsymbol{\delta}}(\boldsymbol{y})\right) \text{ for all } \boldsymbol{\theta},\boldsymbol{\delta} \quad (11)$$

with $\log \pi^{(s)}(\boldsymbol{y})$ being the log-evidence of $\boldsymbol{y}|\boldsymbol{s} \sim p(\boldsymbol{y}|\boldsymbol{s})$ and the improper but sparsity-promoting prior $t(\boldsymbol{s})$, whereas $\log p_{\boldsymbol{\theta},\boldsymbol{\delta}}(\boldsymbol{y})$ is implicitly defined by (7). Consequently, we can interpret the maximization of $\log p_{\boldsymbol{\theta},\boldsymbol{\delta}}(\boldsymbol{y})$ over $(\boldsymbol{\theta},\boldsymbol{\delta})$ as the evidence maximization over the space of variational approximations to a model with a sparsity-inducing and regularizing prior. However, contrary to classical SBL, we have the additional degree of freedom to choose the specifics of $\boldsymbol{\gamma}_{\boldsymbol{\theta}}(\boldsymbol{z})$ and $p_{\boldsymbol{\delta}}(\boldsymbol{z})$ without any constraint while maintaining the connection to sparsity.

## 3.2 Compressive Sensing VAE

The integral version of $p_{\boldsymbol{\theta},\boldsymbol{\delta}}(\boldsymbol{s})$ in (10) strongly resembles the VAE's decomposition of $p(\boldsymbol{x})$ in (5). In fact, by constraining $p_{\boldsymbol{\delta}}(\boldsymbol{z}) = p(\boldsymbol{z}) = \mathcal{N}(\boldsymbol{z};\boldsymbol{0},\mathbf{I})$, the distribution $p_{\boldsymbol{\theta},\boldsymbol{\delta}}(\boldsymbol{s}) = p_{\boldsymbol{\theta}}(\boldsymbol{s})$ corresponds to a subset of all possible distributions covered by the VAE decomposition in (5). This motivates to choose $p_{\boldsymbol{\delta}}(\boldsymbol{z})$ in this exact manner and realize $\boldsymbol{\gamma}_{\boldsymbol{\theta}}(\boldsymbol{z})$ as the output of a NN decoder. Generally, we aim to optimize the log-evidence $\sum_i \log p_{\boldsymbol{\theta}}(\boldsymbol{y}_i)$ in (11) with respect to $\boldsymbol{\theta}$ solely given compressed and noisy observations $\mathcal{Y}$. However, equivalent to VAEs, the log-evidence $\sum_i \log p_{\boldsymbol{\theta}}(\boldsymbol{y}_i)$ is intractable for $p(\boldsymbol{z}) = \mathcal{N}(\boldsymbol{z};\boldsymbol{0},\mathbf{I})$, which is why a tractable lower bound has to be derived. A tractable lower bound on the log-evidence of a statistical model with latent variables is obtained by subtracting the KL divergence between the conditional distribution of the latent variables given the observations

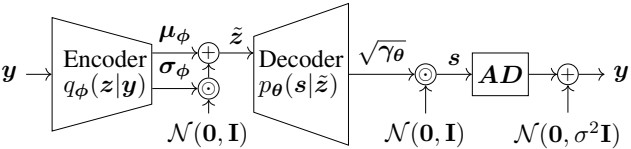

Figure 1: A schematic of the sparsity-inducing CSVAE.

and a tractable variational distribution from the log-evidence (cf. (6)). For the model in (7) and i.i.d. training data, this results in

$$\sum_{\boldsymbol{y}_i \in \mathcal{Y}} \log p_{\boldsymbol{\theta}}(\boldsymbol{y}_i) \geq L_{(\boldsymbol{\theta},\boldsymbol{\phi})}^{(\text{CSVAE})} = \sum_{\boldsymbol{y}_i \in \mathcal{Y}} \log p_{\boldsymbol{\theta}}(\boldsymbol{y}_i) - \mathrm{D}_{\text{KL}}(q_{\boldsymbol{\phi}}(\boldsymbol{z},\boldsymbol{s}|\boldsymbol{y}_i)||p_{\boldsymbol{\theta}}(\boldsymbol{z},\boldsymbol{s}|\boldsymbol{y}_i)) \tag{12}$$

$$= \sum_{\boldsymbol{y}_i \in \mathcal{Y}} \mathbb{E}_{q_{\boldsymbol{\phi}}(\boldsymbol{z},\boldsymbol{s}|\boldsymbol{y}_i)} \left[ \log \frac{p(\boldsymbol{y}_i|\boldsymbol{s})p_{\boldsymbol{\theta}}(\boldsymbol{s}|\boldsymbol{z})p(\boldsymbol{z})}{q_{\boldsymbol{\phi}}(\boldsymbol{z},\boldsymbol{s}|\boldsymbol{y}_i)} \right], \tag{13}$$

where the derivation of (13) from (12) is given in Appendix B. Based on insights from the training principle explained in Section 2.4, the variational posterior $q_{\boldsymbol{\phi}}(\boldsymbol{z},\boldsymbol{s}|\boldsymbol{y}_i)$ in (13) can be simplified to $q_{\boldsymbol{\phi}}(\boldsymbol{z},\boldsymbol{s}|\boldsymbol{y}_i) = p_{\boldsymbol{\theta}}(\boldsymbol{s}|\boldsymbol{z},\boldsymbol{y}_i)q_{\boldsymbol{\phi}}(\boldsymbol{z}|\boldsymbol{y}_i)$ with closed-form $p_{\boldsymbol{\theta}}(\boldsymbol{s}|\boldsymbol{z},\boldsymbol{y}_i)$ and only requires variational parameters $\boldsymbol{\phi}$ for $q_{\boldsymbol{\phi}}(\boldsymbol{z}|\boldsymbol{y}_i)$ to be tractable. As a result, the adapted ELBO $L_{(\boldsymbol{\theta},\boldsymbol{\phi})}^{(\text{CSVAE})}$ equals

$$\sum_{\boldsymbol{y}_i \in \mathcal{Y}} \mathbb{E}_{q_{\boldsymbol{\phi}}(\boldsymbol{z},\boldsymbol{s}|\boldsymbol{y}_i)} \left[ \log \frac{p(\boldsymbol{y}_i|\boldsymbol{s})p_{\boldsymbol{\theta}}(\boldsymbol{s}|\boldsymbol{z})p(\boldsymbol{z})}{p_{\boldsymbol{\theta}}(\boldsymbol{s}|\boldsymbol{z},\boldsymbol{y}_i)q_{\boldsymbol{\phi}}(\boldsymbol{z}|\boldsymbol{y}_i)} \right] = \sum_{\boldsymbol{y}_i \in \mathcal{Y}} \Big( \mathbb{E}_{q_{\boldsymbol{\phi}}(\boldsymbol{z}|\boldsymbol{y}_i)} \Big[ \mathbb{E}_{p_{\boldsymbol{\theta}}(\boldsymbol{s}|\boldsymbol{z},\boldsymbol{y}_i)} \big[ \log p_{\boldsymbol{\theta}}(\boldsymbol{y}_i|\boldsymbol{s}) \big]$$

$$- \mathrm{D}_{\text{KL}}(p_{\boldsymbol{\theta}}(\boldsymbol{s}|\boldsymbol{z},\boldsymbol{y}_i)||p_{\boldsymbol{\theta}}(\boldsymbol{s}|\boldsymbol{z})) \Big] - \mathrm{D}_{\text{KL}}(q_{\boldsymbol{\phi}}(\boldsymbol{z}|\boldsymbol{y}_i)||p(\boldsymbol{z})) \Big). \tag{14}$$

To ensure the training of standard VAEs to be tractable, $\mathbb{E}_{q_{\boldsymbol{\phi}}(\boldsymbol{z}|\boldsymbol{y}_i)}[\cdot]$ is generally approximated by a single-sample Monte-Carlo estimation [37]. We build on the same strategy and approximate (14) by

$$L_{(\boldsymbol{\theta},\boldsymbol{\phi})}^{(\text{CSVAE})} \approx \sum_{\boldsymbol{y}_i \in \mathcal{Y}} \Big( \mathbb{E}_{p_{\boldsymbol{\theta}}(\boldsymbol{s}|\tilde{\boldsymbol{z}}_i,\boldsymbol{y}_i)}[\log p(\boldsymbol{y}_i|\boldsymbol{s})] - \mathrm{D}_{\text{KL}}(q_{\boldsymbol{\phi}}(\boldsymbol{z}|\boldsymbol{y}_i)||p(\boldsymbol{z})) - \mathrm{D}_{\text{KL}}(p_{\boldsymbol{\theta}}(\boldsymbol{s}|\tilde{\boldsymbol{z}}_i,\boldsymbol{y}_i)||p_{\boldsymbol{\theta}}(\boldsymbol{s}|\tilde{\boldsymbol{z}}_i)) \Big) \tag{15}$$

with $\tilde{\boldsymbol{z}}_i \sim q_{\boldsymbol{\phi}}(\boldsymbol{z}|\boldsymbol{y}_i)$ and $\mathbb{E}_{p_{\boldsymbol{\theta}}(\boldsymbol{s}|\tilde{\boldsymbol{z}}_i,\boldsymbol{y}_i)}[\log p(\boldsymbol{y}_i|\boldsymbol{s})]$ exhibiting a closed-form solution detailed in Appendix D. In addition, $\mathrm{D}_{\text{KL}}(p_{\boldsymbol{\theta}}(\boldsymbol{s}|\tilde{\boldsymbol{z}}_i,\boldsymbol{y}_i)||p_{\boldsymbol{\theta}}(\boldsymbol{s}|\tilde{\boldsymbol{z}}_i))$ is the tractable KL divergence of two multivariate Gaussians, and $\mathrm{D}_{\text{KL}}(q_{\boldsymbol{\phi}}(\boldsymbol{z}|\boldsymbol{y}_i)||p(\boldsymbol{z}))$ matches the KL divergence from standard VAEs (cf. [37]). Both are specified in Appendix E. The adapted ELBO in (15) resembles the classical ELBO of ordinary VAEs with a modified reconstruction loss and an additional KL divergence. Equivalent to ordinary VAEs, we model $q_{\boldsymbol{\phi}}(\boldsymbol{z}|\boldsymbol{y}_i) = \mathcal{N}(\boldsymbol{z};\boldsymbol{\mu}_{\boldsymbol{\phi}}(\boldsymbol{y}_i),\text{diag}(\boldsymbol{\sigma}_{\boldsymbol{\phi}}^2(\boldsymbol{y}_i)))$ and realize $(\boldsymbol{\mu}_{\boldsymbol{\phi}}(\cdot),\boldsymbol{\sigma}_{\boldsymbol{\phi}}^2(\cdot))$ by a NN encoder, and $\boldsymbol{\gamma}_{\boldsymbol{\theta}}(\cdot)$ by a NN decoder. The adapted ELBO in (15) forms a differentiable and variational approximation of the model's log-evidence $\sum_i \log p_{\boldsymbol{\theta}}(\boldsymbol{y}_i)$. Based on Theorem 3.1 and (11), it holds that

$$\sum_{\boldsymbol{y}_i \in \mathcal{Y}} \log \pi^{(s)}(\boldsymbol{y}_i) \geq \sum_{\boldsymbol{y}_i \in \mathcal{Y}} \log p_{\boldsymbol{\theta}}(\boldsymbol{y}_i) \geq L_{(\boldsymbol{\theta},\boldsymbol{\phi})}^{(\text{CSVAE})} \text{ for all } \boldsymbol{\theta},\boldsymbol{\phi}. \tag{16}$$

Thus, $L_{(\boldsymbol{\theta},\boldsymbol{\phi})}^{(\text{CSVAE})}$ serves as a variational lower bound of a sparsity-inducing log-evidence. We denote the resulting VAE as Compressive Sensing VAE (CSVAE), and its schematic is presented in Fig. 1. It resembles the vanilla VAE with the main difference that the encoder takes a compressed observation $\boldsymbol{y}$ as input and the decoder solely outputs conditional variances $\boldsymbol{\gamma}_{\boldsymbol{\theta}}(\tilde{\boldsymbol{z}})$ of $\boldsymbol{s}|\boldsymbol{z}$. In case of varying measurement matrices $\boldsymbol{A}_i$ for each training, validation and test sample, we use the least-squares estimate $\hat{\boldsymbol{x}}_i^{(\text{LS})} = \boldsymbol{A}_i^{\text{T}}(\boldsymbol{A}_i\boldsymbol{A}_i^{\text{T}})^{-1}\boldsymbol{y}_i$ instead of $\boldsymbol{y}_i$ as encoder input. The CSVAE's training with ground-truth data and pseudo-code for training are outlined in Appendix H and N (cf. algorithm 1).

### 3.3 Compressive Sensing GMM

By considering discrete and finite $p_{\boldsymbol{\delta}}(\boldsymbol{z})$ in (7), i.e., $p_{\boldsymbol{\delta}}(\boldsymbol{z}=k) = p_{\boldsymbol{\delta}}(k) = \rho_k$, we observe that the resulting $p_{\boldsymbol{\theta},\boldsymbol{\delta}}(\boldsymbol{s})$ strongly resembles the GMM's decomposition of $p(\boldsymbol{x})$ in (4). This motivates to choose $p_{\boldsymbol{\delta}}(\boldsymbol{z})$ in exactly this manner and $p_{\boldsymbol{\theta}}(\boldsymbol{s}|\boldsymbol{z}) = p(\boldsymbol{s}|k) = \mathcal{N}(\boldsymbol{s};\boldsymbol{0},\text{diag}(\boldsymbol{\gamma}_k))$, i.e., $\boldsymbol{s}$ is modelled as GMM with zero means and diagonal covariance matrices. Equivalent to CSVAEs in Section 3.2,

the distribution $p(s|k, y)$ contained in the model's posterior $p(k, s|y) = p(s|k, y)p(k|y)$ exhibits a closed-form solution (cf. (8)). Moreover, due to the linear relation between $y$ and $s$ in (7) and the assumption that $s|k$ is Gaussian, $y|k$ is also Gaussian with zero mean and covariance matrix

$$C_k^{y|k} = AD\text{diag}(\gamma_k)D^TA^T + \sigma^2 I. \tag{17}$$

and $p(k|y)$ is computable in closed form by Bayes, i.e., $p(k|y) = (p(y|k)p(k))/(\sum_k p(y|k)p(k))$. Thus, $p(k, s|y)$ exhibits a closed form, and an EM algorithm can be applied, whose e-step coincides with computing $p(k, s|y)$. Based on the extended EM algorithm in [23], where the authors fit a vanilla GMM (4) in the pixel domain using compressed image patches, we formulate an m-step tailored to our model (7). We denote the resulting model as Compressive Sensing GMM (CSGMM). The m-step aims to update the CSGMM's parameters $\{\rho_k, \gamma_k\}$ by optimizing $\sum_i \mathbb{E}_{p_t(k, s|x_i)}[\log p(x_i, s, k)]$, whose closed-form solution is given in the following lemma.

**Lemma 3.2.** *Let the statistical model in (7) be given with discrete $p_\delta(z = k) = p(k) = \rho_k$ and $\gamma_\theta(z) = \gamma_k$, and let $\mathcal{Y}$ contain i.i.d. noisy and compressed observations $y_i$. Given $p_t(k|y_i)$ and $p_t(s|k, y_i) = \mathcal{N}(s; \mu_{k,t}^{s|y_i,k}, C_{k,t}^{s|y_i,k})$ in the tth iteration of an EM algorithm, the updates of $\{\rho_k, \gamma_k\}$*

$$\gamma_{k,(t+1)} = \frac{\sum_{y_i \in \mathcal{Y}} p_t(k|y_i)(|\mu_{k,t}^{s|y_i,k}|^2 + diag(C_{k,t}^{s|y_i,k}))}{\sum_{y_i \in \mathcal{Y}} p_t(k|y_i)}, \; \rho_{k,(t+1)} = \frac{\sum_{y_i \in \mathcal{Y}} p_t(k|y_i)}{|\mathcal{Y}|} \tag{18}$$

*form a valid m-step and, thus, guarantee to improve the log-evidence in every iteration.*

The proof of Lemma 3.2, the CSGMM's training with ground-truth data $\mathcal{S}$ and pseudo-code are outlined in Appendix F, H and N (cf. algorithm 4), respectively.

### 3.4  Discussion

From a broader perspective, our proposed algorithm represents two consecutive stages:

1) Choose a dictionary $D$, with respect to which a general signal set $\mathcal{C}$ of interest is compressible (e.g. a wavelet basis for the set of natural images).

2) Learn the statistical characteristics of the specific signal subset $\mathcal{S} \subset \mathcal{C}$ of interest in its compressible domain (e.g. the subset of handwritten digits).

Due to the strong regularization of stage 1) constraining the search space in stage 2), our model can learn from a few corrupted samples $\mathcal{Y}$. To ensure the search space to be within the set of compressible signals with respect to $D$, we enforce $s|z$ in (7) and, thus, $s$ to have zero mean (cf. Theorem 3.1).

**Limitation.**    On the one hand, this restriction regularizes the problem at hand. On the other hand, it generally introduces a bias, which potentially prevents perfectly learning the unknown distribution of $s$ as it is not always possible to decompose a distribution in this way. As a result, the proposed model is biased towards capturing the sparsity-specific features from $\mathcal{Y}$ and being an effective prior for (1) rather than learning a comprehensive representation of the true $p(s)$.

**Distinction from generative model-based CS.**    Generative model-based CS builds on a generator $G_\theta : z \mapsto x = G_\theta(z)$ (e.g., a GAN or VAE with $z \in \mathbb{R}^F$, $x \in \mathbb{R}^N$ and $F \ll N$ [4]. Arguably, the notion of compressibility is still included in this setup, since it assumes that every $x$ can be perfectly encoded by only a few $F$ values. Moreover, these models typically assign a simplistic fixed distribution, e.g. $p(z) = \mathcal{N}(z; 0, I)$, to their compressible domain. Consequently, while generative model-based CS replaces the dictionary with a learnable mapping $G_\theta$ and constrains the compressible domain by forcing it to be representable by, e.g., $\mathcal{N}(0, I)$, our approach does the exact opposite. We keep the mapping from the compressible into the original domain fixed by a dictionary $D$ and learn a non-trivial statistical model $p_{\theta,\delta}(s)$ in the compressible domain.

**Connection to uncertainty quantification.**    While coming from different motivations, our approach shares similarities with methods from uncertainty quantification, namely so-called prior networks [39]. There, the general idea is to let the NN output the parameters of a conditioned conjugate prior (conditioned on the NN input) instead of directly the quantities of interest (cf. [40, 41]). As a result, one can use entropy and mutual information measures to quantify different types of uncertainty. Equivalently, our proposed models output the parameters $\gamma_\theta(z)$ of a conditioned conjugated prior $p_\theta(s|z)$, which is why they enable to quantify their uncertainty by the same measures.

**Computational complexity.** It is noteworthy that our algorithm's inference after training requires no optimization algorithm but only consists of, e.g., a feed-forward operation through a comparably small VAE. Thus, our proposed method's signal reconstruction comes with computational benefits and differs in this context from other approaches. On the downside, naively implementing the training requires computing and storing the posterior covariance matrices in (8), which can lead to a non-negligible computational and memory-related overhead for very high dimensions. To overcome this issue, we derive equivalent reformulations of the CSVAE's and CSGMM's update steps for training that circumvent the explicit computation of these matrices. More precisely, the reformulated update steps solely require the explicit storing and inversion of the observations' covariance matrices (cf. (17)), which are typically much lower dimensional. The reformulations are given in Appendix I.

## 4 Experiments

### 4.1 Experimental Setup

**Datasets.** We evaluate the performance on four datasets. We use the MNIST dataset ($N = 784$) for evaluation [42, (CC-BY-SA 3.0 license)]. Moreover, we use an artificial 1D dataset of piecewise smooth functions of dimension $N = 256$. For that, we combine *HeaviSine* functions with polynomials of quadratic degree and randomly placed discontinuities, which are frequently used for CS and compressible in the wavelet domain [43, 17]. The generation and examples are outlined in Appendix J. We also use a dataset of $64 \times 64$ cropped celebA images ($N = 3 \cdot 64^2 = 12288$) [44][3] and evaluate on the FashionMNIST dataset ($N = 784$) in Appendix L [45, (MIT license)].

**Measurement matrix & evaluation metric.** For the simulations on piecewise smooth functions, we use a separate measurement matrix $\boldsymbol{A}_i$ for each training, validation and test sample, where all $\boldsymbol{A}_i$ contain i.i.d. Gaussian entries $\boldsymbol{A}_{i,kl} \sim \mathcal{N}(0, \frac{1}{M})$. For all remaining simulations, we use one fixed measurement matrix $\boldsymbol{A}$ each with equally distributed i.i.d. Gaussian entries for the whole dataset. We evaluate the distortion performance by a normalized MSE nMSE $= 1/N_{\text{test}} \sum_{i=1}^{N_{\text{test}}} (\|\hat{\boldsymbol{x}}_i - \boldsymbol{x}_i\|_2^2 / N)$ with $\hat{\boldsymbol{x}}_i$ being the estimation of $\boldsymbol{x}_i$ and the SSIM [46] with $N_{\text{test}}$ being set to $5000$ in any simulation.

**Baselines & hyperparameters.** As non-learnable baselines, we use Lasso [12] and SBL [19], where we either adjust Lasso's shrinkage parameter on a ground-truth dataset or use the configurations from [4]. As baselines, which can learn from compressed data, we use CSGAN [30] and CKSVD [24] with OMP, sparsity level 4 and 288 learnable dictionary atoms. We solely evaluate CKSVD on piecewise smooth functions due to its requirement of varying measurement matrices for observing the training samples. For CSGAN on MNIST, we use the configuration specified in [30].[4] In any simulation, we use $K = 32$ components for the proposed CSGMM and the proposed CSVAE's en- and decoders contain two fully-connected layers with ReLU activation and one following linear layer, respectively. We utilize Adam for optimization [47], and only consider the CME approximation of the estimators (cf. Section 2.4) since we did not observe notable differences to the alternative estimators in Appendix G. We utilize an overcomplete Daubechies *db4* dictionary [48] in all estimators with fixed dictionaries. For all estimators except CSGAN, all images and estimates are normalized and clipped between $0$ and $1$, respectively. For CSGAN, images are normalized between $-1$ and $1$ following [4, 30], and for evaluation, test images and their estimations are then re-normalized between $0$ and $1$. For a more detailed overview of hyperparameter configurations, see Appendix K.

### 4.2 Results

**Reconstruction.** In Fig. 2 and 3, the reconstruction performance of all estimators for MNIST, the piecewise smooth functions, and celebA is shown. All trainable models are trained on compressed training samples $\mathcal{Y}$ of dimension $M$ without any ground-truth information during training (cf. Section 2.1). The models are then used to estimate test signals $\boldsymbol{x}_i$ from observations of the same dimension $M$ (cf. (1)) for evaluation. In the case of the piecewise smooth functions, we also add noise of a 10dB signal-to-noise ratio (SNR) for training and testing, i.e., $\sigma_n^2 = \mathbb{E}[\|\boldsymbol{A}\boldsymbol{x}\|_2^2]/(M \cdot 10)$ in (1).

---

[3]The celebA dataset is released under a custom license for non-commercial research use.

[4]The original work of CSGAN [30] does not provide results for varying measurement matrices as well as for celebA trained on solely compressed data. We also did not find a working hyperparameter configuration, so we leave out CSGAN for the piecewise smooth functions and the celebA dataset.

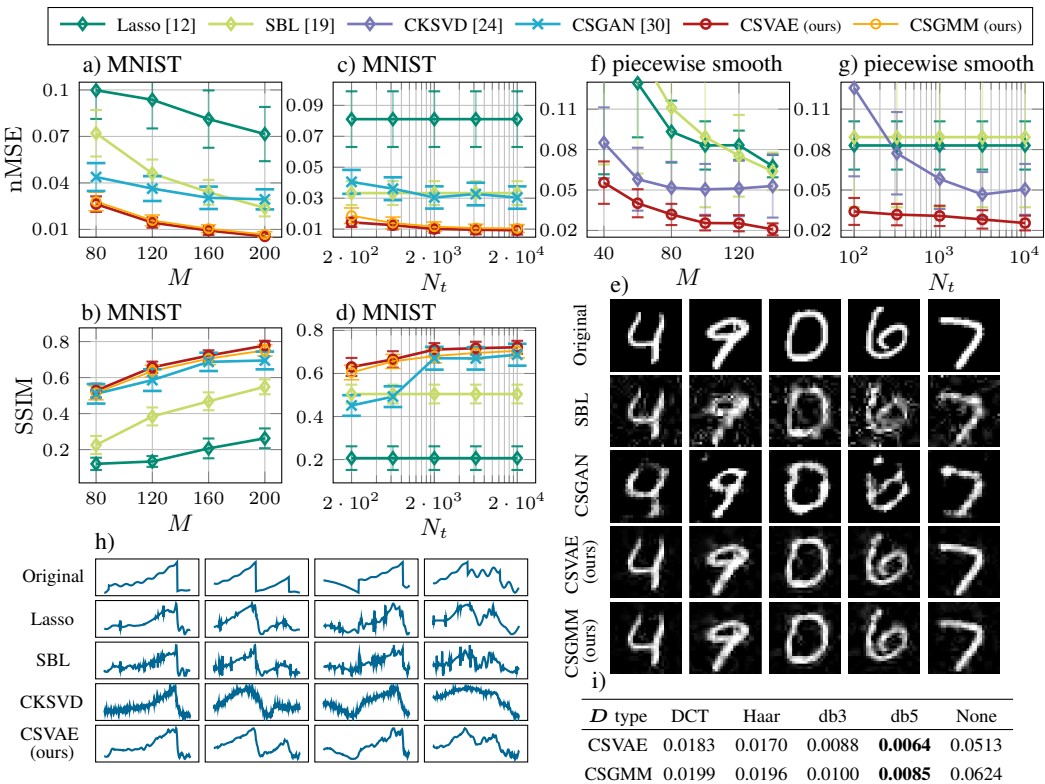

Figure 2: a) and b) nMSE and SSIM over $M$ ($N_t = 20000$, MNIST), c) and d) nMSE and SSIM over $N_t$ ($M = 160$, MNIST), e) exemplary reconstructed MNIST images ($M = 200$, $N_t = 20000$), f) nMSE over $M$ ($\text{SNR}_{\text{dB}} = 10\text{dB}$, $N_t = 10000$, piece-wise smooth fct.), g) nMSE over $N_t$ ($\text{SNR}_{\text{dB}} = 10\text{dB}$, $M = 100$, piece-wise smooth fct.), h) exemplary reconstructed piece-wise smooth fct. ($M = 100$, $N_t = 1000$), i) nMSE comparison of dictionaries (MNIST, $M = 160$, $N_t = 20000$).

In Fig. 2 a)-d), the nMSE and SSIM on MNIST is shown for a) and b) varying observation dimensions $M$ and the fixed number $N_t = 20000$ of training samples, and c) and d) vice versa with fixed $M = 160$. The error bars represent standard deviations. In terms of both distortion metrics nMSE and SSIM, CSVAE and CSGMM perform overall the best. In Fig. 2 c), exemplary reconstructed MNIST images for $M = 200$ and $N_t = 20000$ are shown. Perceptually, CSGAN emphasizes different reconstruction aspects than the proposed CSVAE and CSGMM. While CSVAE and CSGMM successfully recover details of the MNIST images, CSGAN prioritizes the similarity in contrast between bright and dark areas. This can be explained by their different estimation strategies. The estimation of CSGANs is restricted to lie on the learned manifold, which has been adjusted based on the training dataset. In consequence, when the test sample $x^*$ contains details that cannot be found in the training dataset, CSGANs cannot reconstruct these but rather output a similar but realistic representative from the manifold. On the contrary, our proposed estimation scheme in Section 2.4 is not restricted to a manifold but rather infers conditional distributions (cf. (8)) over the whole linear space. Moreover, Fig. 2 c) and d) demonstrate that the proposed model can effectively learn from only a few hundred compressed and noisy samples. In Fig. 2 i), CSVAE and CSGMM are compared using different dictionaries types. The overall nMSE performance remains good for different dictionaries except for applying the models directly in the pixel domain.

In Fig. 2 f), g) and h), the nMSE over varying $M$ (see f)), varying $N_t$ (see g)) and exemplary reconstructed samples (see h)) are shown for the set of piecewise smooth functions. Here, we only evaluated the proposed CSVAE.[5] Compared to the baselines Lasso, SBL, and CKSVD, the CSVAE performs the best in distortion (see f) and g)) as well as perception (see h)). In Fig. 3 a)-e), the same plots are shown for the celebA dataset. Despite the significantly larger dimension, the results are consistent with those for MNIST and the set of piecewise smooth functions.

---

[5]We leave out the CSGMM due to the need to compute separate posterior covariance matrices for all training samples by using varying measurement matrices (cf. (8)), which leads to a computational overhead.

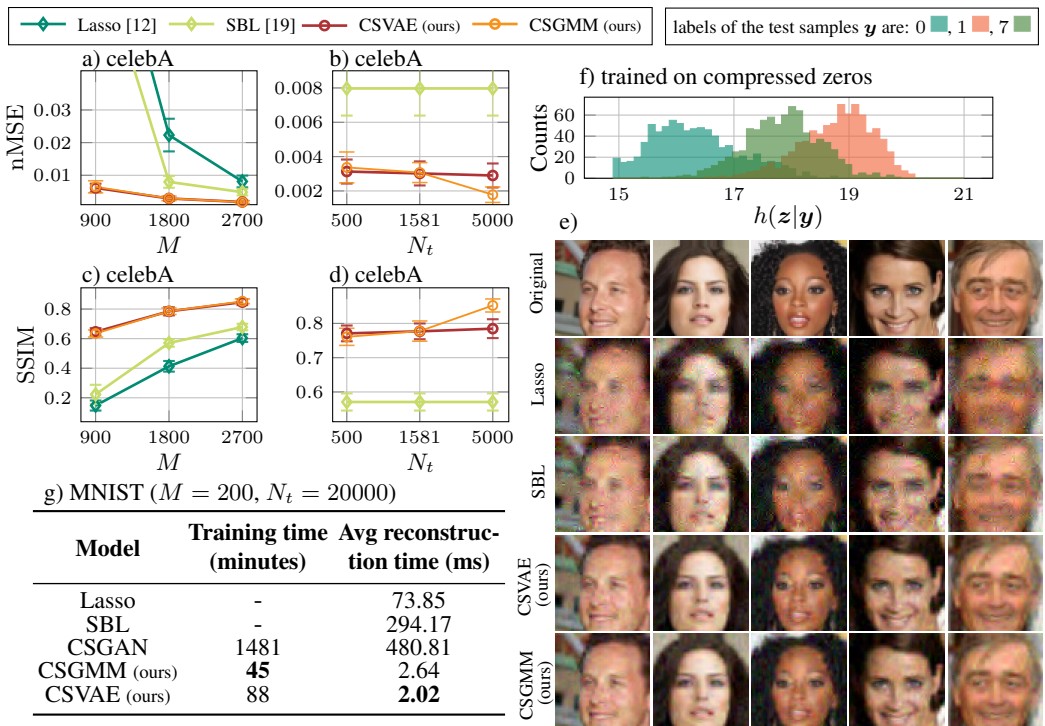

Figure 3: a) and b) nMSE and SSIM over $M$ ($N_t = 5000$), c) and d) nMSE and SSIM over $N_t$ ($M = 1800$), e) exemplary reconstructed celebA images ($M = 2700$, $N_t = 5000$), f) histogram of $h(\boldsymbol{z}|\boldsymbol{y})$ for compressed test MNIST images of digits 0,1 and 7, where the CSVAE is trained on compressed zeros, g) training and reconstruction time for MNIST ($M = 200$, $N_t = 20000$).

**Uncertainty quantification.** One possibility to determine the model's uncertainty is to determine the differential entropy $h(\boldsymbol{z}|\boldsymbol{y})$ of the CSVAE's encoder distribution $q_\phi(\boldsymbol{z}|\boldsymbol{y})$. Since $q_\phi(\boldsymbol{z}|\boldsymbol{y})$ is a Gaussian with diagonal covariance matrix (cf. Section 3.2), $h(\boldsymbol{z}|\boldsymbol{y})$ can be calculated efficiently in closed form. In Fig. 3 f), a histogram is shown with values of $h(\boldsymbol{z}|\boldsymbol{y})$. The CSVAE is trained on solely compressed MNIST zeros. We then forward compressed zeros, ones and sevens and evaluate $h(\boldsymbol{z}|\boldsymbol{y})$. It can be seen that the CSVAE's encoder identifies the observations that are distinct to the training dataset by providing larger entropy values on average for the compressed ones and sevens.

**Runtime.** Fig. 3 g) shows our measured training time as well as average reconstruction time of the MAP-based estimators (cf. Section G) for MNIST, $M = 200$, $N_t = 20000$. CSGMM and CSVAE are considerably faster than the baselines validating the corresponding discussion in Section 3.4.

**Additional results.** The results in Fig. 2 and 3 display the performance for CSGAN and the proposed CSVAE and CSGMM trained on solely compressed data. However, all three models can also learn from ground-truth data (cf. Appendix H). In Appendix L, we include this comparison for MNIST, provide results on FashionMNIST, analyze the estimators' robustness, and plot further reconstructions for all datasets. Moreover, in Appendix M, we analyze the runtime for training and reconstruction in more detail and give an overview of the used compute resources.

## 5 Conclusion

In this work, we introduced a new type of learnable prior for regularizing ill-conditioned inverse problems denoted by sparse Bayesian generative modeling. Our approach shares the property of classical CS methods of utilizing compressibility, but at the same time, it incorporates the adaptability to training data. Due to its strong regularization towards sparsity, it can learn from a few corrupted data samples. It applies to any type of compressible signal and can be used for uncertainty quantification. While this work focused on setting up the sparse Bayesian generative modeling framework, extensions to learnable dictionaries, circumventing the inversion of the observations' covariance matrices during training, and perception-emphasizing estimators are part of future work.

## Acknowledgments and Disclosure of Funding

This research was supported by Rohde & Schwarz GmbH & Co. KG. The authors gratefully acknowledge the financial support and resources provided by the company. The authors also sincerely appreciate the valuable discussions with Dominik Semmler and Benedikt Fesl.

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

## A  Proof of Theorem 3.1

To prove Theorem 3.1, we restate (3), i.e., there exists a $C > 0$ such that for all $\boldsymbol{\gamma} > 0$ and $\boldsymbol{s}$

$$\mathcal{N}(\boldsymbol{s}; \boldsymbol{0}, \mathrm{diag}(\boldsymbol{\gamma})) \leq t(\boldsymbol{s}) = C \cdot \prod_{i=1}^{N} \frac{1}{|s_i|}. \tag{19}$$

Based on (19), there exists a $C > 0$ such that for all $\boldsymbol{\theta}, \boldsymbol{s}, \boldsymbol{z}$ with $\boldsymbol{\gamma_\theta}(\boldsymbol{z}) > \boldsymbol{0}$

$$p_{\boldsymbol{\theta}}(\boldsymbol{s}|\boldsymbol{z}) = \mathcal{N}(\boldsymbol{s}; \boldsymbol{0}, \mathrm{diag}(\boldsymbol{\gamma_\theta}(\boldsymbol{z}))) \leq t(\boldsymbol{s}) = C \cdot \prod_{i=1}^{N} \frac{1}{|s_i|}. \tag{20}$$

By multiplying both sides in (20) with $p_{\boldsymbol{\delta}}(\boldsymbol{z})$ and integrating over $\boldsymbol{z}$, we conclude that there exists a $C > 0$ such that for all $\boldsymbol{\theta}, \boldsymbol{s}, \boldsymbol{\delta}$ with $\boldsymbol{\gamma_\theta}(\boldsymbol{z}) > \boldsymbol{0}$

$$p_{\boldsymbol{\theta},\boldsymbol{\delta}}(\boldsymbol{s}) = \int p_{\boldsymbol{\delta}}(\boldsymbol{z}) p_{\boldsymbol{\theta}}(\boldsymbol{s}|\boldsymbol{z}) \mathrm{d}\boldsymbol{z} \leq \int p_{\boldsymbol{\delta}}(\boldsymbol{z}) t(\boldsymbol{s}) \mathrm{d}\boldsymbol{z} = t(\boldsymbol{s}) \tag{21}$$

independent of the particular choice of $p_{\boldsymbol{\delta}}(\boldsymbol{z})$. In case of discrete and finite $p_{\boldsymbol{\delta}}(\boldsymbol{z})$ the integral corresponds to a summation.

## B  Derivation of the adapted ELBO for CSVAEs

We write out the KL divergence and utilize that $p_{\boldsymbol{\theta}}(\boldsymbol{y})$ is independent of the expectation, i.e.,

$$L_{(\boldsymbol{\theta},\boldsymbol{\phi})}^{(\mathrm{CSVAE})} = \sum_{\boldsymbol{y}_i \in \mathcal{Y}} \log p_{\boldsymbol{\theta}}(\boldsymbol{y}_i) - \mathrm{D_{KL}}(q_{\boldsymbol{\phi}}(\boldsymbol{z},\boldsymbol{s}|\boldsymbol{y}_i) || p_{\boldsymbol{\theta}}(\boldsymbol{z},\boldsymbol{s}|\boldsymbol{y}_i)) \tag{22}$$

$$= \sum_{\boldsymbol{y}_i \in \mathcal{Y}} \mathbb{E}_{q_{\boldsymbol{\phi}}(\boldsymbol{z},\boldsymbol{s}|\boldsymbol{y}_i)} \left[ \log \frac{p_{\boldsymbol{\theta}}(\boldsymbol{y}_i) p_{\boldsymbol{\theta}}(\boldsymbol{z},\boldsymbol{s}|\boldsymbol{y}_i)}{q_{\boldsymbol{\phi}}(\boldsymbol{z},\boldsymbol{s}|\boldsymbol{y}_i)} \right] = \sum_{\boldsymbol{y}_i \in \mathcal{Y}} \mathbb{E}_{q_{\boldsymbol{\phi}}(\boldsymbol{z},\boldsymbol{s}|\boldsymbol{y}_i)} \left[ \log \frac{p(\boldsymbol{y}_i|\boldsymbol{s}) p_{\boldsymbol{\theta}}(\boldsymbol{s}|\boldsymbol{z}) p(\boldsymbol{z})}{q_{\boldsymbol{\phi}}(\boldsymbol{z},\boldsymbol{s}|\boldsymbol{y}_i)} \right]. \tag{23}$$

## C  Conditional Posterior in the Noise-Free Case

In the noise-free case, the closed-form covariance matrix and mean of $p_{\boldsymbol{\theta}}(\boldsymbol{s}|\boldsymbol{z},\boldsymbol{y})$ are given by

$$\boldsymbol{C}_{\boldsymbol{\theta}}^{\boldsymbol{s}|\boldsymbol{y},\boldsymbol{z}}(\boldsymbol{z}) = \left( \mathbf{I} - \mathrm{diag}(\sqrt{\boldsymbol{\gamma_\theta}(\boldsymbol{z})}) \left( \boldsymbol{AD}\mathrm{diag}(\sqrt{\boldsymbol{\gamma_\theta}(\boldsymbol{z})}) \right)^{\dagger} \boldsymbol{AD} \right) \mathrm{diag}(\boldsymbol{\gamma_\theta}(\boldsymbol{z})) \tag{24}$$

$$\boldsymbol{\mu}_{\boldsymbol{\theta}}^{\boldsymbol{s}|\boldsymbol{y},\boldsymbol{z}}(\boldsymbol{z}) = \mathrm{diag}(\sqrt{\boldsymbol{\gamma_\theta}(\boldsymbol{z})}) \left( \boldsymbol{AD}\mathrm{diag}(\sqrt{\boldsymbol{\gamma_\theta}(\boldsymbol{z})}) \right)^{\dagger} \boldsymbol{y}. \tag{25}$$

with $(\cdot)^{\dagger}$ being the Moore-Penrose inverse [19].

## D  Closed-Form Solution of the CSVAE Reconstruction Loss

$$\mathbb{E}_{p_{\boldsymbol{\theta}}(\boldsymbol{s}|\tilde{\boldsymbol{z}}_i,\boldsymbol{y}_i)}[\log p(\boldsymbol{y}_i|\boldsymbol{s})] = \mathbb{E}_{p_{\boldsymbol{\theta}}(\boldsymbol{s}|\tilde{\boldsymbol{z}}_i,\boldsymbol{y}_i)} \left[ -\frac{1}{2}\left( M\log(2\pi\sigma^2) + \frac{1}{\sigma^2}\|\boldsymbol{y}_i - \boldsymbol{AD}\boldsymbol{s}\|_2^2 \right) \right] \tag{26}$$

$$= -\frac{1}{2}\left( M\log(2\pi\sigma^2) + \frac{1}{\sigma^2}\mathbb{E}_{p_{\boldsymbol{\theta}}(\boldsymbol{s}|\tilde{\boldsymbol{z}}_i,\boldsymbol{y}_i)} \left[ \|\boldsymbol{y}_i\|_2^2 - \boldsymbol{y}_i^{\mathrm{T}}\boldsymbol{AD}\boldsymbol{s} \right. \right. \tag{27}$$
$$\left. \left. - \boldsymbol{s}^{\mathrm{T}}\boldsymbol{D}^{\mathrm{T}}\boldsymbol{A}^{\mathrm{T}}\boldsymbol{y}_i + \mathrm{tr}(\boldsymbol{AD}\boldsymbol{s}\boldsymbol{s}^{\mathrm{T}}\boldsymbol{D}^{\mathrm{T}}\boldsymbol{A}^{\mathrm{T}}) \right] \right)$$

$$= -\frac{1}{2}\left( M\log(2\pi\sigma^2) + \frac{1}{\sigma^2}\left( \|\boldsymbol{y}_i\|_2^2 - \boldsymbol{y}_i^{\mathrm{T}}\boldsymbol{AD}\boldsymbol{\mu}_{\boldsymbol{\theta}}^{\boldsymbol{s}|\boldsymbol{y}_i,\tilde{\boldsymbol{z}}_i}(\tilde{\boldsymbol{z}}_i) - \boldsymbol{\mu}_{\boldsymbol{\theta}}^{\boldsymbol{s}|\boldsymbol{y}_i,\tilde{\boldsymbol{z}}_i}(\tilde{\boldsymbol{z}}_i)^{\mathrm{T}}\boldsymbol{D}^{\mathrm{T}}\boldsymbol{A}^{\mathrm{T}}\boldsymbol{y}_i \right. \right. \tag{28}$$
$$\left. \left. + \mathrm{tr}\left( \boldsymbol{AD}\left( \boldsymbol{C}_{\boldsymbol{\theta}}^{\boldsymbol{s}|\boldsymbol{y}_i,\tilde{\boldsymbol{z}}_i}(\tilde{\boldsymbol{z}}_i) + \boldsymbol{\mu}_{\boldsymbol{\theta}}^{\boldsymbol{s}|\boldsymbol{y}_i,\tilde{\boldsymbol{z}}_i}(\tilde{\boldsymbol{z}}_i)\boldsymbol{\mu}_{\boldsymbol{\theta}}^{\boldsymbol{s}|\boldsymbol{y}_i,\tilde{\boldsymbol{z}}_i}(\tilde{\boldsymbol{z}}_i)^{\mathrm{T}} \right) \boldsymbol{D}^{\mathrm{T}}\boldsymbol{A}^{\mathrm{T}} \right) \right) \right)$$

$$= -\frac{1}{2}\left( M\log(2\pi\sigma^2) + \frac{1}{\sigma^2}\left( \|\boldsymbol{y}_i - \boldsymbol{AD}\boldsymbol{\mu}_{\boldsymbol{\theta}}^{\boldsymbol{s}|\boldsymbol{y}_i,\tilde{\boldsymbol{z}}_i}(\tilde{\boldsymbol{z}}_i)\|_2^2 + \mathrm{tr}(\boldsymbol{AD}\boldsymbol{C}_{\boldsymbol{\theta}}^{\boldsymbol{s}|\boldsymbol{y}_i,\tilde{\boldsymbol{z}}_i}(\tilde{\boldsymbol{z}}_i)\boldsymbol{D}^{\mathrm{T}}\boldsymbol{A}^{\mathrm{T}}) \right) \right) \tag{29}$$

where $\boldsymbol{\mu}_{\boldsymbol{\theta}}^{\boldsymbol{s}|\boldsymbol{y}_i,\tilde{\boldsymbol{z}}_i}(\tilde{\boldsymbol{z}}_i)$ and $\boldsymbol{C}_{\boldsymbol{\theta}}^{\boldsymbol{s}|\boldsymbol{y}_i,\tilde{\boldsymbol{z}}_i}(\tilde{\boldsymbol{z}}_i)$ are the mean and covariance matrix of $p_{\boldsymbol{\theta}}(\boldsymbol{s}|\tilde{\boldsymbol{z}}_i,\boldsymbol{y}_i)$.

# E    Closed-Form Solution of the CSVAE KL Divergences

By inserting the closed form of the KL divergence between two Gaussians (cf. [49]), we get

$$
\begin{aligned}
D_{\mathrm{KL}}(p_{\boldsymbol{\theta}}(\boldsymbol{s}|\tilde{\boldsymbol{z}}_i,\boldsymbol{y}_i)||p_{\boldsymbol{\theta}}(\boldsymbol{s}|\tilde{\boldsymbol{z}}_i)) = \frac{1}{2}\Big( &\log\det\left(\mathrm{diag}(\boldsymbol{\gamma}_{\boldsymbol{\theta}}(\tilde{\boldsymbol{z}}_i))\right) - \log\det\left(\boldsymbol{C}_{\boldsymbol{\theta}}^{\boldsymbol{s}|\boldsymbol{y}_i,\tilde{\boldsymbol{z}}_i}(\tilde{\boldsymbol{z}}_i)\right) - S \\
&+ \mathrm{tr}\left(\mathrm{diag}\left((\boldsymbol{\gamma}_{\boldsymbol{\theta}}(\tilde{\boldsymbol{z}}_i)^{-1})\,\boldsymbol{C}_{\boldsymbol{\theta}}^{\boldsymbol{s}|\boldsymbol{y}_i,\tilde{\boldsymbol{z}}_i}(\tilde{\boldsymbol{z}}_i)\right) + \boldsymbol{\mu}_{\boldsymbol{\theta}}^{\boldsymbol{s}|\boldsymbol{y}_i,\tilde{\boldsymbol{z}}_i}(\tilde{\boldsymbol{z}}_i)^{\mathrm{T}}\mathrm{diag}\left(\boldsymbol{\gamma}_{\boldsymbol{\theta}}(\tilde{\boldsymbol{z}}_i)^{-1}\right)\boldsymbol{\mu}_{\boldsymbol{\theta}}^{\boldsymbol{s}|\boldsymbol{y}_i,\tilde{\boldsymbol{z}}_i}(\tilde{\boldsymbol{z}}_i)\right)
\end{aligned}
\tag{30}
$$

where $\boldsymbol{\mu}_{\boldsymbol{\theta}}^{\boldsymbol{s}|\boldsymbol{y}_i,\tilde{\boldsymbol{z}}_i}(\tilde{\boldsymbol{z}}_i)$ and $\boldsymbol{C}_{\boldsymbol{\theta}}^{\boldsymbol{s}|\boldsymbol{y}_i,\tilde{\boldsymbol{z}}_i}(\tilde{\boldsymbol{z}}_i)$ are the mean and covariance matrix of $p_{\boldsymbol{\theta}}(\boldsymbol{s}|\tilde{\boldsymbol{z}}_i,\boldsymbol{y}_i)$. Moreover [47],

$$
D_{\mathrm{KL}}(q_{\boldsymbol{\phi}}(\boldsymbol{z}|\boldsymbol{y}_i)||p(\boldsymbol{z})) = -\frac{1}{2}\sum_{j=1}^{N_L}(1+\log\sigma_{j,\boldsymbol{\phi}}^2(\boldsymbol{y}_i)) - \mu_{j,\boldsymbol{\phi}}(\boldsymbol{y}_i) - \sigma_{j,\boldsymbol{\phi}}^2(\boldsymbol{y}_i))
\tag{31}
$$

with $q_{\boldsymbol{\phi}}(\boldsymbol{z}|\boldsymbol{y}_i) = \mathcal{N}(\boldsymbol{z};\boldsymbol{\mu}_{\boldsymbol{\phi}}(\boldsymbol{y}_i),\mathrm{diag}(\boldsymbol{\sigma}_{\boldsymbol{\phi}}^2(\boldsymbol{y}_i)))$ and $\mu_{j,\boldsymbol{\phi}}(\boldsymbol{y}_i)$ and $\sigma_{j,\boldsymbol{\phi}}^2(\boldsymbol{y}_i))$ being the $j$th entry of $\boldsymbol{\mu}_{\boldsymbol{\phi}}(\boldsymbol{y}_i)$ and $\boldsymbol{\sigma}_{\boldsymbol{\phi}}^2(\boldsymbol{y}_i)$, respectively. Additionally, $N_L$ denotes the CSVAE's latent dimension.

# F    Proof of Lemma 3.2

The optimization problem we aim to solve is given by

$$
\{\rho_{k,(t+1)},\boldsymbol{\gamma}_{k,(t+1)}\} = \underset{\{\rho_k,\boldsymbol{\gamma}_k\}}{\operatorname{argmax}} \sum_{\boldsymbol{y}_i\in\mathcal{Y}} \mathbb{E}_{p_t(k,\boldsymbol{s}|\boldsymbol{y}_i)}\left[\log p(\boldsymbol{y}_i,\boldsymbol{s},k)\right] \quad \text{s.t.} \sum_k \rho_k = 1.
\tag{32}
$$

First, we reformulate the objective as

$$
\sum_{\boldsymbol{y}_i\in\mathcal{Y}} \mathbb{E}_{p_t(k,\boldsymbol{s}|\boldsymbol{y}_i)}\left[\log p(\boldsymbol{y}_i,\boldsymbol{s},k)\right] = \sum_{\boldsymbol{y}_i\in\mathcal{Y}} \mathbb{E}_{p_t(k,\boldsymbol{s}|\boldsymbol{y}_i)}\left[\log p(\boldsymbol{y}_i|\boldsymbol{s}) + \log p(\boldsymbol{s}|k) + \log p(k)\right],
\tag{33}
$$

Moreover, we observe that $\log p(\boldsymbol{y}_i|\boldsymbol{s})$ does not depend on $\{\rho_k,\boldsymbol{\gamma}_k\}$ and we can leave it out from the optimization problem. Additionally,

$$
\sum_{\boldsymbol{y}_i\in\mathcal{Y}} \mathbb{E}_{p_t(k|\boldsymbol{y}_i)}\left[\mathbb{E}_{p_t(\boldsymbol{s}|\boldsymbol{y}_i,k)}\left[\log p(\boldsymbol{s}|k) + \log p(k)\right]\right]
$$

$$
= \sum_{\boldsymbol{y}_i\in\mathcal{Y}}\sum_{k=1}^{K} p_t(k|\boldsymbol{y}_i)\left(-\frac{1}{2}\left(S\log 2\pi + \sum_{j=1}^{S}\log\gamma_{k,j} + \sum_{j=1}^{S}\frac{\mathbb{E}_{p_t(\boldsymbol{s}|\boldsymbol{y}_i,k)}\left[|s_j|^2\right]}{\gamma_{k,j}}\right) + \log\rho_k\right).
\tag{34}
$$

where we inserted $p(\boldsymbol{s}|k) = \mathcal{N}(\boldsymbol{s};\boldsymbol{0},\mathrm{diag}(\boldsymbol{\gamma}_k))$ (cf. (7)) and $\gamma_{k,j}$ denotes the $j$th entry of $\boldsymbol{\gamma}_k$. In the next step, let $\mathbb{E}_{p_t(\boldsymbol{s}|\boldsymbol{y}_i,k)}\left[|s_j|^2\right] = |\mu_{k,j}^{\boldsymbol{s}|\boldsymbol{y}_i,k}|^2 + C_{k,j,j}^{\boldsymbol{s}|\boldsymbol{y}_i,k}$, where $\mu_{k,j}^{\boldsymbol{s}|\boldsymbol{y}_i,k}$ and $C_{k,j,j}^{\boldsymbol{s}|\boldsymbol{y}_i,k}$ denote the $j$th entry of $\boldsymbol{\mu}_{k,t}^{\boldsymbol{s}|\boldsymbol{y}_i,k}$ and the diagonal of $\boldsymbol{C}_{k,t}^{\boldsymbol{s}|\boldsymbol{y}_i,k}$ in the $t$th iteration, respectively. Thus, our optimization problem of interest can be restated as

$$
\underset{\{\rho_k,\boldsymbol{\gamma}_k\}}{\operatorname{argmax}} \sum_{\boldsymbol{y}_i\in\mathcal{Y}}\sum_{k=1}^{K} p_t(k|\boldsymbol{y}_i)\left(-\frac{1}{2}\left(S\log 2\pi + \sum_{j=1}^{S}\left(\log\gamma_{k,j} + \frac{|\mu_{k,j}^{\boldsymbol{s}|\boldsymbol{y}_i,k}|^2 + C_{k,j,j}^{\boldsymbol{s}|\boldsymbol{y}_i,k}}{\gamma_{k,j}}\right)\right) + \log\rho_k\right)
$$

$$
\text{s.t.} \sum_k \rho_k = 1
\tag{35}
$$

with Lagrangian

$$
\mathcal{L} = \sum_{\boldsymbol{y}_i\in\mathcal{Y}}\sum_{k=1}^{K} p_t(k|\boldsymbol{y}_i)\Big(-\frac{1}{2}\Big(S\log 2\pi + \sum_{j=1}^{S}\Big(\log\gamma_{k,j} + \frac{|\mu_{k,j}^{\boldsymbol{s}|\boldsymbol{y}_i,k}|^2 + C_{k,j,j}^{\boldsymbol{s}|\boldsymbol{y}_i,k}}{\gamma_{k,j}}\Big)\Big) +
$$

$$
\log\rho_k\Big) + \nu(1-\sum_k \rho_k)
\tag{36}
$$

and Lagrangian multiplier $\nu$. Taking the derivative of $\mathcal{L}$ with respect to $\gamma_{m,q}$ and $\rho_m$ and setting it to zero leads to

$$\frac{\partial}{\partial \gamma_{m,q}} \mathcal{L} = -\frac{1}{2} \sum_{\boldsymbol{y}_i \in \mathcal{Y}} p_t(m|\boldsymbol{y}_i) \left( \frac{1}{\gamma_{m,q}} - \frac{|\mu_{m,q}^{\boldsymbol{s}|\boldsymbol{y}_i,k}|^2 + \boldsymbol{C}_{m,q,q}^{\boldsymbol{s}|\boldsymbol{y}_i,k}}{\gamma_{m,q}^2} \right) = 0 \tag{37}$$

$$\frac{\partial}{\partial \rho_m} \mathcal{L} = \sum_{\boldsymbol{y}_i \in \mathcal{Y}} \frac{p_t(m|\boldsymbol{y}_i)}{\rho_m} - \nu = 0 \tag{38}$$

and, thus,

$$\gamma_{m,q,(t+1)} = \frac{\sum_{\boldsymbol{y}_i \in \mathcal{Y}} p_t(m|\boldsymbol{y}_i) \left( |\mu_{m,q}^{\boldsymbol{s}|\boldsymbol{y}_i,k}|^2 + \boldsymbol{C}_{m,q,q}^{\boldsymbol{s}|\boldsymbol{y}_i,k} \right)}{\sum_{\boldsymbol{y}_i \in \mathcal{Y}} p_t(m|\boldsymbol{y}_i)} \tag{39}$$

$$\rho_{m,(t+1)} = \frac{\sum_{\boldsymbol{y}_i \in \mathcal{Y}} p_t(m|\boldsymbol{y}_i)}{\nu} \tag{40}$$

$$\nu = |\mathcal{Y}| \tag{41}$$

where (41) comes from the normalization of $\sum_k \rho_k = 1$.

## G  Estimators Based on the CSVAE and CSGMM

**CSVAE.** The CME approximation in (9) with the proposed CSVAE in Section 3.2 is given by

$$\hat{\boldsymbol{x}}_{\text{CSVAE,CME}}^* = \frac{\boldsymbol{D}}{|\mathcal{Z}|} \sum_{\tilde{\boldsymbol{z}}_i \in \mathcal{Z}} \mathbb{E}_{p_{\boldsymbol{\theta}}(\boldsymbol{s}|\tilde{\boldsymbol{z}}_i, \boldsymbol{y})} \left[ \boldsymbol{s}|\boldsymbol{y}, \tilde{\boldsymbol{z}}_i \right] \tag{42}$$

with $\mathcal{Z}$ containing samples $\tilde{\boldsymbol{z}}_i \sim q_{\boldsymbol{\phi}}(\boldsymbol{z}|\boldsymbol{y})$. Alternatively, one can estimate $\boldsymbol{x}^*$ based on a newly observed $\boldsymbol{y}$ in the following way. We use the mean $\boldsymbol{\mu}_{\boldsymbol{\phi}}(\boldsymbol{y})$ as maximum a posteriori (MAP) estimate based on $q_{\boldsymbol{\phi}}(\boldsymbol{z}|\boldsymbol{y})$ and, then, estimate $\boldsymbol{x}^*$ by

$$\hat{\boldsymbol{x}}_{\text{CSVAE,MAP}}^* = \boldsymbol{D} \, \mathbb{E}_{p_{\boldsymbol{\theta}}(\boldsymbol{s}|\boldsymbol{\mu}_{\boldsymbol{\phi}}(\boldsymbol{y}), \boldsymbol{y})} \left[ \boldsymbol{s}|\boldsymbol{y}, \boldsymbol{\mu}_{\boldsymbol{\phi}}(\boldsymbol{y}) \right] = \boldsymbol{D} \boldsymbol{\mu}_{\boldsymbol{\theta}}^{\boldsymbol{s}|\boldsymbol{y}, \boldsymbol{\mu}_{\boldsymbol{\phi}}(\boldsymbol{y})} (\boldsymbol{\mu}_{\boldsymbol{\phi}}(\boldsymbol{y})). \tag{43}$$

This method is applied in [33] to reduce computational complexity, but also deviates from the CME approximation and potentially improves the perceptual quality of the reconstruction.

**CSGMM.** In case of the proposed CSGMM in Section 3.3 the CME approximation in (9) exhibits a closed form, i.e.,

$$\hat{\boldsymbol{x}}_{\text{CSGMM,CME}}^* = \boldsymbol{D} \sum_k p(k|\boldsymbol{y}) \, \mathbb{E}_{p(\boldsymbol{s}|k,\boldsymbol{y})}[\boldsymbol{s}|k, \boldsymbol{y}]. \tag{44}$$

An alternative estimator of $\boldsymbol{x}^*$ is given by

$$\hat{\boldsymbol{x}}_{\text{CSGMM,MAP}}^* = \boldsymbol{D} \, \mathbb{E}_{p(\boldsymbol{s}|\hat{k}_{\text{MAP}}, \boldsymbol{y})}[\boldsymbol{s}|\hat{k}_{\text{MAP}}, \boldsymbol{y}] \tag{45}$$

with $\hat{k}_{\text{MAP}} = \arg\max p(k|\boldsymbol{y})$ [23]. A pseudo-code for all estimators is provided in Appendix N (cf. algorithm 2-6).

## H  CSVAE and CSGMM Training on Ground-Truth Data

The CSGMM as well as the CSVAE can both be trained given ground-truth datasets. For training the CSGMM, this dataset either contains the compressible ground-truth signals $\boldsymbol{s}_i$, i.e., $\mathcal{S}$, or the signals $\boldsymbol{x}_i$ themself, i.e., $\mathcal{X}$. On the other hand, for training the CSVAE, both datasets must also contain their corresponding observations $\boldsymbol{y}_i$. Thus, the training dataset must be $\mathcal{G}$ or $\mathcal{W}$ (cf. Section 2.1).

**CSGMM.** To train the CSGMM, the training goal is to maximize the log-evidence of the training dataset. In case of having access to $\mathcal{S}$, the vanilla EM algorithm is employed with the modification of enforcing the GMM's means to be zero and covariance matrices to be diagonal in every update

step. More precisely, after computing the posteriors $p_t(k|\boldsymbol{s}_i)$ by the Bayes rule in the $t$th iteration, the m-step is given by

$$\gamma_{k,(t+1)} = \frac{\sum_{\boldsymbol{s}_i \in \mathcal{S}} p_t(k|\boldsymbol{s}_i)|\boldsymbol{s}_i|^2}{\sum_{\boldsymbol{s}_i \in \mathcal{S}} p_t(k|\boldsymbol{s}_i)}, \ \rho_{k,(t+1)} = \frac{\sum_{\boldsymbol{s}_i \in \mathcal{S}} p_t(k|\boldsymbol{s}_i)}{|\mathcal{S}|}. \tag{46}$$

In case of having access to $\mathcal{X}$, the same modified EM algorithm from Section 3.3 is employed by exchanging $\boldsymbol{A}\boldsymbol{D}$ with $\boldsymbol{D}$ and setting the noise variance $\sigma^2$ to some small value.

**CSVAE.** To train the CSVAE using the dataset $\mathcal{G}$, we also aim optimize the log-evidence of the training dataset

$$\sum_{(\boldsymbol{s}_i,\boldsymbol{y}_i) \in \mathcal{G}} \log p_{\boldsymbol{\theta}}(\boldsymbol{s}_i, \boldsymbol{y}_i) = \sum_{(\boldsymbol{s}_i,\boldsymbol{y}_i) \in \mathcal{G}} \log p_{\boldsymbol{\theta}}(\boldsymbol{y}_i|\boldsymbol{s}_i) + \log p_{\boldsymbol{\theta}}(\boldsymbol{s}_i) \tag{47}$$

By considering that $\log p_{\boldsymbol{\theta}}(\boldsymbol{y}_i|\boldsymbol{s}_i)$ does not depend on the CSVAE's parameters (see (7)), introducing additional variational parameters $\boldsymbol{\phi}$ and approximating $p_{\boldsymbol{\theta}}(\boldsymbol{z}|\boldsymbol{y}, \boldsymbol{s})$ with $q_{\boldsymbol{\phi}}(\boldsymbol{z}|\boldsymbol{y})$, we apply the standard reformulations of VAEs, which ends up in an ELBO resembling the standard ELBO in (6), given by

$$\tilde{L}(\boldsymbol{\theta}, \boldsymbol{\phi}) = \sum_{(\boldsymbol{s}_i,\boldsymbol{y}_i) \in \mathcal{G}} \log p_{\boldsymbol{\theta}}(\boldsymbol{s}_i|\tilde{\boldsymbol{z}}_i) - \mathrm{D_{KL}}(q_{\boldsymbol{\phi}}(\boldsymbol{z}|\boldsymbol{y}_i)||p(\boldsymbol{z})) \tag{48}$$

with $\tilde{\boldsymbol{z}}_i \sim q_{\boldsymbol{\phi}}(\boldsymbol{z}|\boldsymbol{y}_i)$ and $p_{\boldsymbol{\theta}}(\boldsymbol{s}_i)$ being defined in (10) with $p(\boldsymbol{z}) = \mathcal{N}(\boldsymbol{z}; \boldsymbol{0}, \mathbf{I})$. In the case of training the CSVAE with $\mathcal{W}$, the same training procedure from Section 3.2 is applied, resulting in the modified ELBO $\tilde{L}_{(\boldsymbol{\theta}, \boldsymbol{\phi})}$ approximated by

$$\sum_{(\boldsymbol{x}_i,\boldsymbol{y}_i) \in \mathcal{W}} \Big( \mathbb{E}_{p_{\boldsymbol{\theta}}(\boldsymbol{s}|\tilde{\boldsymbol{z}}_i, \boldsymbol{x}_i)}[\log p(\boldsymbol{x}_i|\boldsymbol{s})] - \mathrm{D_{KL}}(q_{\boldsymbol{\phi}}(\boldsymbol{z}|\boldsymbol{y}_i)||p(\boldsymbol{z})) - \mathrm{D_{KL}}(p_{\boldsymbol{\theta}}(\boldsymbol{s}|\tilde{\boldsymbol{z}}_i, \boldsymbol{x}_i)||p_{\boldsymbol{\theta}}(\boldsymbol{s}|\tilde{\boldsymbol{z}}_i)) \Big) \tag{49}$$

where we set the variational distribution $q_{\boldsymbol{\phi}}(\boldsymbol{s}, \boldsymbol{z}|\boldsymbol{x}_i, \boldsymbol{y}_i)$ of the latent variables given the observations to $q_{\boldsymbol{\phi}}(\boldsymbol{z}|\boldsymbol{y}_i)p_{\boldsymbol{\theta}}(\boldsymbol{s}|\boldsymbol{z}, \boldsymbol{x}_i)$. Consequently, instead of computing the posterior $p_{\boldsymbol{\theta}}(\boldsymbol{s}|\boldsymbol{z}, \boldsymbol{y}_i)$, we use $p_{\boldsymbol{\theta}}(\boldsymbol{s}|\boldsymbol{z}, \boldsymbol{x}_i)$, which is given by (8) with $\boldsymbol{D}$ instead of $\boldsymbol{A}\boldsymbol{D}$ and setting the noise variance $\sigma^2$ to some small value. Additionally, to compute $\mathbb{E}_{p_{\boldsymbol{\theta}}(\boldsymbol{s}|\tilde{\boldsymbol{z}}_i, \boldsymbol{x}_i)}[\log p(\boldsymbol{x}_i|\boldsymbol{s})]$ we replace $\boldsymbol{y}_i$, $M$ and $\boldsymbol{A}\boldsymbol{D}$ in (29) with $\boldsymbol{x}_i$, $N$ and $\boldsymbol{D}$, respectively.

# I  Implementation Aspects for Reducing the Computational Overhead

In a first step towards a computationally more efficient implementation for training the CSVAE and CSGMM, we reformulate the expressions of the conditional mean $\boldsymbol{\mu}_{\boldsymbol{\theta}}^{s|y,z}(\boldsymbol{z})$ and conditional covariance matrix $\boldsymbol{C}_{\boldsymbol{\theta}}^{s|y,z}(\boldsymbol{z})$ from [18, 19] in (8). To do so, we observe that conditioned on $\boldsymbol{z}$, $\boldsymbol{s}$ and $\boldsymbol{y}$ in (7) are jointly Gaussian. In consequence, we can alternatively apply the standard formulas for computing the moments of a conditional distribution for a jointly Gaussian setup [50], i.e.,

$$\boldsymbol{\mu}_{\boldsymbol{\theta}}^{s|y,z}(\boldsymbol{z}) = \boldsymbol{C}_{\boldsymbol{\theta}}^{s,y|z}(\boldsymbol{z}) \left( \boldsymbol{C}_{\boldsymbol{\theta}}^{y|z}(\boldsymbol{z}) \right)^{-1} \boldsymbol{y} \tag{50}$$

$$\boldsymbol{C}_{\boldsymbol{\theta}}^{s|y,z}(\boldsymbol{z}) = \mathrm{diag}(\boldsymbol{\gamma}_{\boldsymbol{\theta}}(\boldsymbol{z})) - \boldsymbol{C}_{\boldsymbol{\theta}}^{s,y|z}(\boldsymbol{z}) \left( \boldsymbol{C}_{\boldsymbol{\theta}}^{y|z}(\boldsymbol{z}) \right)^{-1} \boldsymbol{C}_{\boldsymbol{\theta}}^{s,y|z}(\boldsymbol{z})^{\mathrm{T}} \tag{51}$$

with

$$\boldsymbol{C}_{\boldsymbol{\theta}}^{y|z}(\boldsymbol{z}) = \boldsymbol{A}\boldsymbol{D}\mathrm{diag}(\boldsymbol{\gamma}_{\boldsymbol{\theta}}(\boldsymbol{z}))\boldsymbol{D}^{\mathrm{T}}\boldsymbol{A}^{\mathrm{T}} + \sigma^2 \mathbf{I} \tag{52}$$

and $\boldsymbol{C}_{\boldsymbol{\theta}}^{s,y|z}(\boldsymbol{z}) = \mathrm{diag}(\boldsymbol{\gamma}_{\boldsymbol{\theta}}(\boldsymbol{z}))\boldsymbol{D}^{\mathrm{T}}\boldsymbol{A}^{\mathrm{T}}$. Importantly, we do not need to explicitly compute $\boldsymbol{C}_{\boldsymbol{\theta}}^{s|y,z}(\boldsymbol{z})$ according to (51). The subsequent reformulations differ between CSGMM and CSVAE.

**CSGMM.** For the CSGMM, only the diagonal entries of $\boldsymbol{C}_{\boldsymbol{\theta}}^{s|y,z}(\boldsymbol{z})$ must be explicitly computed, i.e.,

$$\mathrm{diag}\left( \boldsymbol{C}_{\boldsymbol{\theta}}^{s|y,z}(\boldsymbol{z}) \right) = \boldsymbol{\gamma}_{\boldsymbol{\theta}}(\boldsymbol{z}) - \mathrm{diag}\left( \boldsymbol{C}_{\boldsymbol{\theta}}^{s,y|z}(\boldsymbol{z}) \left( \boldsymbol{C}_{\boldsymbol{\theta}}^{y|z}(\boldsymbol{z}) \right)^{-1} \boldsymbol{C}_{\boldsymbol{\theta}}^{s,y|z}(\boldsymbol{z})^{\mathrm{T}} \right). \tag{53}$$

The matrix $C_{\boldsymbol{\theta}}^{\boldsymbol{s},\boldsymbol{y}|\boldsymbol{z}}(\boldsymbol{z})\left(C_{\boldsymbol{\theta}}^{\boldsymbol{y}|\boldsymbol{z}}(\boldsymbol{z})\right)^{-1}$ (i.e., $C_k^{\boldsymbol{s},\boldsymbol{y}|k}\left(C_k^{\boldsymbol{y}|k}\right)^{-1}$[6]) has been precomputed for the conditional mean in (50) and subsequently determining $\mathrm{diag}\left(C_k^{\boldsymbol{s},\boldsymbol{y}|k}\left(C_k^{\boldsymbol{y}|k}\right)^{-1}\left(C_k^{\boldsymbol{s},\boldsymbol{y}|k}\right)^{\mathrm{T}}\right)$ only requires $\mathcal{O}(SM)$ operations. Moreover, $\left(C_k^{\boldsymbol{y}|k}\right)^{-1}$ has already been determined for evaluating the posterior distributions $p(k|\boldsymbol{y})$ in the preceding e-step (cf. Section 3.3). The closed-form m-step in (18) only takes $\mathrm{diag}\left(C_k^{\boldsymbol{s}|\boldsymbol{y},k}\right)$ for which we can directly use (53). In this way, we circumvent explicitly computing the full posterior covariance matrices in (8), rendering training the CSGMM more efficient.

**CSVAE.** The objective for training the CSVAE is given in (15), which consists of the modified reconstruction loss in (29) as well as the KL divergences in (30) and (31). While $\boldsymbol{\mu}_{\boldsymbol{\theta}}^{\boldsymbol{s}|\boldsymbol{y},\boldsymbol{z}}(\boldsymbol{z})$ in (29) can be directly computed using (50), we reformulate the trace-term in (29) to circumvent the explicit computation of $C_{\boldsymbol{\theta}}^{\boldsymbol{s}|\boldsymbol{y},\boldsymbol{z}}(\boldsymbol{z})$. To do so, we apply the following steps

$$\mathrm{tr}(\boldsymbol{A}\boldsymbol{D}C_{\boldsymbol{\theta}}^{\boldsymbol{s}|\boldsymbol{y}_i,\tilde{\boldsymbol{z}}_i}(\tilde{\boldsymbol{z}}_i)\boldsymbol{D}^{\mathrm{T}}\boldsymbol{A}^{\mathrm{T}}) = \mathrm{tr}\left(C_{\boldsymbol{\theta}}^{\boldsymbol{s}|\boldsymbol{y}_i,\tilde{\boldsymbol{z}}_i}(\tilde{\boldsymbol{z}}_i)\boldsymbol{D}^{\mathrm{T}}\boldsymbol{A}^{\mathrm{T}}\boldsymbol{A}\boldsymbol{D}\right) = \tag{54}$$

$$\sigma^2\,\mathrm{tr}\left(C_{\boldsymbol{\theta}}^{\boldsymbol{s}|\boldsymbol{y}_i,\tilde{\boldsymbol{z}}_i}(\tilde{\boldsymbol{z}}_i)\left(\left(C_{\boldsymbol{\theta}}^{\boldsymbol{s}|\boldsymbol{y}_i,\tilde{\boldsymbol{z}}_i}(\tilde{\boldsymbol{z}}_i)\right)^{-1} - \mathrm{diag}(\boldsymbol{\gamma}_{\boldsymbol{\theta}}^{-1}(\tilde{\boldsymbol{z}}_i))\right)\right) = \sigma^2\left(S - \sum_{j=1}^{S}\frac{C_{\boldsymbol{\theta},j,j}^{\boldsymbol{s}|\boldsymbol{y}_i,\tilde{\boldsymbol{z}}_i}(\tilde{\boldsymbol{z}}_i)}{\boldsymbol{\gamma}_{\boldsymbol{\theta},j}(\tilde{\boldsymbol{z}}_i)}\right) \tag{55}$$

with $C_{\boldsymbol{\theta},j,j}^{\boldsymbol{s}|\boldsymbol{y}_i,\tilde{\boldsymbol{z}}_i}(\tilde{\boldsymbol{z}}_i)$ being the $j$th diagonal entry of $C_{\boldsymbol{\theta}}^{\boldsymbol{s}|\boldsymbol{y}_i,\tilde{\boldsymbol{z}}_i}(\tilde{\boldsymbol{z}}_i)$ and $\boldsymbol{\gamma}_{\boldsymbol{\theta},j}(\tilde{\boldsymbol{z}}_i)$ being the $j$th entry of $\boldsymbol{\gamma}_{\boldsymbol{\theta}}(\tilde{\boldsymbol{z}}_i)$. For the derivation, we mainly apply the formula of $C_{\boldsymbol{\theta}}^{\boldsymbol{s}|\boldsymbol{y}_i,\tilde{\boldsymbol{z}}_i}(\tilde{\boldsymbol{z}}_i)$ in (8). By observing that the $\sigma^2$ in (55) cancels out with the $1/\sigma^2$ in (29), the trace-term in (29) cancels out with $-S + \mathrm{tr}\left(\mathrm{diag}\left((\boldsymbol{\gamma}_{\boldsymbol{\theta}}(\tilde{\boldsymbol{z}}_i)^{-1})\,C_{\boldsymbol{\theta}}^{\boldsymbol{s}|\boldsymbol{y}_i,\tilde{\boldsymbol{z}}_i}(\tilde{\boldsymbol{z}}_i)\right)\right)$ in (30). We also reformulate $\log\det\left(C_{\boldsymbol{\theta}}^{\boldsymbol{s}|\boldsymbol{y}_i,\tilde{\boldsymbol{z}}_i}(\tilde{\boldsymbol{z}}_i)\right)$ in (30), rendering the explicit computation of $C_{\boldsymbol{\theta}}^{\boldsymbol{s}|\boldsymbol{y}_i,\tilde{\boldsymbol{z}}_i}(\tilde{\boldsymbol{z}}_i)$ obsolete, i.e.,

$$\log\det\left(C_{\boldsymbol{\theta}}^{\boldsymbol{s}|\boldsymbol{y}_i,\tilde{\boldsymbol{z}}_i}(\tilde{\boldsymbol{z}}_i)\right) = -\log\det\left(\frac{1}{\sigma^2}\boldsymbol{D}^{\mathrm{T}}\boldsymbol{A}^{\mathrm{T}}\boldsymbol{A}\boldsymbol{D} + \mathrm{diag}(\boldsymbol{\gamma}_{\boldsymbol{\theta}}^{-1}(\tilde{\boldsymbol{z}}_i))\right) = \tag{56}$$

$$-\log\det\left(\left(\frac{1}{\sigma^2}\boldsymbol{D}^{\mathrm{T}}\boldsymbol{A}^{\mathrm{T}}\boldsymbol{A}\boldsymbol{D} + \mathrm{diag}(\boldsymbol{\gamma}_{\boldsymbol{\theta}}^{-1}(\tilde{\boldsymbol{z}}_i))\right)\mathrm{diag}(\boldsymbol{\gamma}_{\boldsymbol{\theta}}(\tilde{\boldsymbol{z}}_i))\mathrm{diag}(\boldsymbol{\gamma}_{\boldsymbol{\theta}}^{-1}(\tilde{\boldsymbol{z}}_i))\right) = \tag{57}$$

$$-\log\det\left(\left(\frac{1}{\sigma^2}\boldsymbol{D}^{\mathrm{T}}\boldsymbol{A}^{\mathrm{T}}\boldsymbol{A}\boldsymbol{D} + \mathrm{diag}(\boldsymbol{\gamma}_{\boldsymbol{\theta}}^{-1}(\tilde{\boldsymbol{z}}_i))\right)\mathrm{diag}(\boldsymbol{\gamma}_{\boldsymbol{\theta}}(\tilde{\boldsymbol{z}}_i))\right) + \log\det(\mathrm{diag}(\boldsymbol{\gamma}_{\boldsymbol{\theta}}(\tilde{\boldsymbol{z}}_i))) = \tag{58}$$

$$-\log\det\left(\frac{1}{\sigma^2}\boldsymbol{D}^{\mathrm{T}}\boldsymbol{A}^{\mathrm{T}}\boldsymbol{A}\boldsymbol{D}\mathrm{diag}(\boldsymbol{\gamma}_{\boldsymbol{\theta}}(\tilde{\boldsymbol{z}}_i)) + \mathbf{I}\right) + \log\det(\mathrm{diag}(\boldsymbol{\gamma}_{\boldsymbol{\theta}}(\tilde{\boldsymbol{z}}_i))) = \tag{59}$$

$$-\log\det\left(\frac{1}{\sigma^2}\boldsymbol{A}\boldsymbol{D}\mathrm{diag}(\boldsymbol{\gamma}_{\boldsymbol{\theta}}(\tilde{\boldsymbol{z}}_i))\boldsymbol{D}^{\mathrm{T}}\boldsymbol{A}^{\mathrm{T}} + \mathbf{I}\right) + \log\det(\mathrm{diag}(\boldsymbol{\gamma}_{\boldsymbol{\theta}}(\tilde{\boldsymbol{z}}_i))) = \tag{60}$$

$$-\log\det(\frac{1}{\sigma^2}\mathbf{I}) - \log\det C_{\boldsymbol{\theta}}^{\boldsymbol{y}|\tilde{\boldsymbol{z}}_i}(\tilde{\boldsymbol{z}}_i) + \log\det(\mathrm{diag}(\boldsymbol{\gamma}_{\boldsymbol{\theta}}(\tilde{\boldsymbol{z}}_i))) = \tag{61}$$

$$M\log\sigma^2 - \log\det C_{\boldsymbol{\theta}}^{\boldsymbol{y}|\tilde{\boldsymbol{z}}_i}(\tilde{\boldsymbol{z}}_i) + \log\det(\mathrm{diag}(\boldsymbol{\gamma}_{\boldsymbol{\theta}}(\tilde{\boldsymbol{z}}_i))) \tag{62}$$

where we use the formula of $C_{\boldsymbol{\theta}}^{\boldsymbol{s}|\boldsymbol{y}_i,\tilde{\boldsymbol{z}}_i}(\tilde{\boldsymbol{z}}_i)$ in (8), $C_{\boldsymbol{\theta}}^{\boldsymbol{y}|\tilde{\boldsymbol{z}}_i}(\tilde{\boldsymbol{z}}_i) = \boldsymbol{A}\boldsymbol{D}\mathrm{diag}(\boldsymbol{\gamma}_{\boldsymbol{\theta}}(\tilde{\boldsymbol{z}}_i))\boldsymbol{D}^{\mathrm{T}}\boldsymbol{A}^{\mathrm{T}} + \sigma^2\mathbf{I}$ as well as Sylvester's determinant theorem for the reformulation from (59) to (60).

In general, equivalent reformulations can also be applied when the CSVAE and CSGMM are trained on ground-truth data (cf. Appendix H) by replacing $C_{\boldsymbol{\theta}}^{\boldsymbol{s}|\boldsymbol{y}_i,\tilde{\boldsymbol{z}}_i}(\tilde{\boldsymbol{z}}_i)$ with $C_{\boldsymbol{\theta}}^{\boldsymbol{s}|\boldsymbol{x}_i,\tilde{\boldsymbol{z}}_i}(\tilde{\boldsymbol{z}}_i)$.

---

[6]For the CSGMM we use the notation from Section 3.3

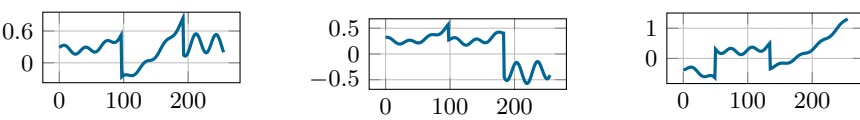

Figure 4: Exemplary signals within the 1D dataset of piecewise smooth functions.

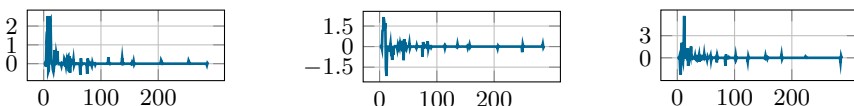

Figure 5: Wavelet Transforms of the exemplary signals in Fig. 4.

## J   Implementation of the 1D Dataset of Piecewise Smooth Functions

The artificial 1D dataset of piecewise smooth functions is generated in the following way:

$$x_i(t) = \begin{cases} \sum_{j=0}^{2} h_{1,i}^{(j)} t^j + a_i^{(1)} \sin(4\pi t + \eta_i^{(1)}), & t \in [0, g_1) \\ \sum_{j=0}^{2} h_{2,i}^{(j)} t^j + a_i^{(2)} \sin(4\pi t + \eta_i^{(2)}), & t \in [g_1, g_2) \\ \sum_{j=0}^{2} h_{3,i}^{(j)} t^j + a_i^{(2)} \sin(4\pi t + \eta_i^{(3)}), & t \in [g_2, 4) \end{cases} \tag{63}$$

where $h_{m,i}^{(j)} \sim \text{Bernoulli}(0.5)$ and either takes $-0.4$ or $0.4$, $a_i^{(m)} \sim \mathcal{N}(0, 0.1^2)$, $\eta_i^{(m)} \sim \mathcal{U}(0, 2\pi)$, $g_1 \sim \mathcal{U}(0, 2)$ and $g_2 \sim \mathcal{U}(2, 4)$. After generation, each $x_i(t)$ is sampled equidistantly $N = 256$ times, and its samples are stored in vectors $\boldsymbol{x}_i$, which then constitute the ground-truth dataset $\mathcal{X}$. Exemplary signals and their wavelet decomposition are given in Fig. 4 and 5, respectively.

## K   Detailed Overview of Hyperparameter Configurations

**MNIST & FashionMNIST.**   The non-learnable baselines for the simulations on MNIST are Lasso as well SBL. We apply Lasso directly in the pixel domain with its shrinkage parameter $\lambda$ set to $0.1$ in line with [4]. For SBL, we use an overcomplete *db4* dictionary with symmetric extension [48]. Moreover, although we do not include noise in the simulations for MNIST, we set $\sigma^2$ in (2) to an increment corresponding to 40dB SNR to be able to apply the computationally efficient reformulations from Appendix I. For CSGAN, we use the exact hyperparameter configurations specified in [30]. For CSGMM, we set the number $K$ of components to $32$ and iterate until the increments of the training dataset's log-evidence reach the tolerance parameter of $10^{-3}$, a standard stopping criterion for GMMs [51]. The CSVAE encoders and decoders contain two fully connected layers with ReLU activation and one following linear layer, respectively. The widths of the layers are set in a way such that for the first two layers, the width increases linearly from the input dimension to $256$, while the final linear layer maps from $256$ to the desired dimension (i.e., either $S$ for the decoder or twice the latent dimension for the encoder). The latent dimension is set to $16$, the learning rate is set to $2 \cdot 10^{-5}$, and the batch size is set to $64$. We use the Adam optimizer for optimization [47]. We once reduce the learning rate by a factor of $2$ during training and stop the training, when the modified ELBO in (15) for a validation set of $5000$ samples does not increase. For CSGMM as well as CSVAE we set $\sigma^2$ in (7) to an increment corresponding to 40dB SNR to be able to apply the training reformulation from Appendix I. We also use the same overcomplete *db4* dictionary as for SBL. Moreover, we use $N_s = 64$ samples to approximate the outer expectation in (9).

**Piecewise smooth function.**   For the simulations on the set of piecewise smooth functions, we adjust the shrinkage parameter of Lasso based on a ground-truth validation dataset of $5000$ samples once for every $M$. Moreover, we also use the overcomplete *db4* dictionary for Lasso as for all other dictionary-based estimators. Instead of choosing $256$, we choose $128$ as the maximum width of the en- and decoder layers. Otherwise, the hyperparameters remain the same as for the simulations on MNIST.

**CelebA.**   For the simulations on celebA, we set the shrinkage parameter $\lambda$ of Lasso to $0.00001$ (cf. [4]). As celebA contains colored images, we choose a block-diagonal dictionary with the overcomplete *db4* dictionary three times along the diagonal and zero matrices in all off-diagonals.

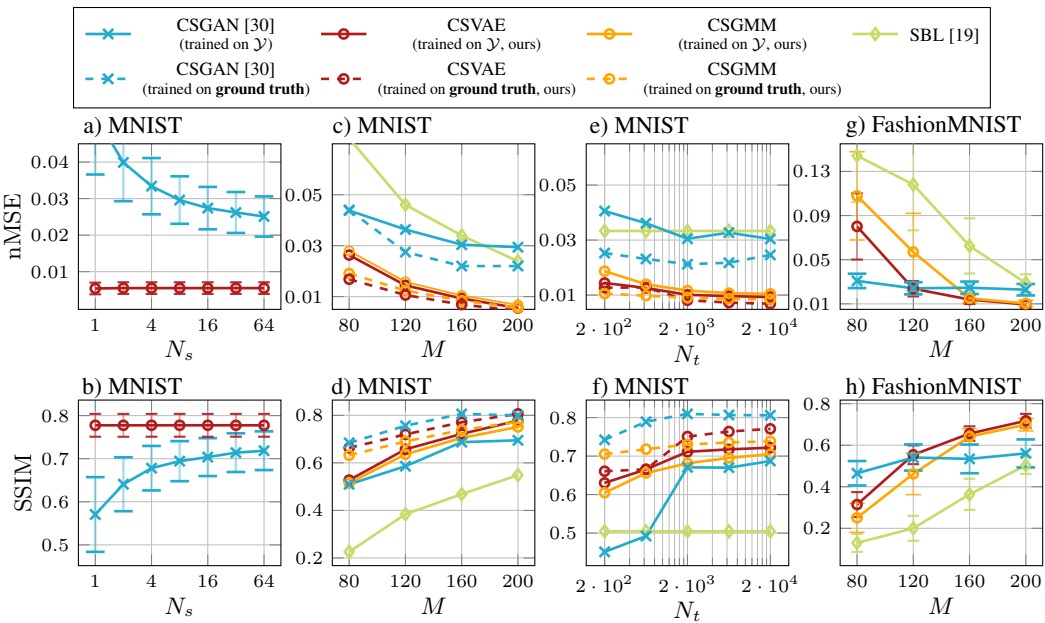

Figure 6: a) and b) nMSE and SSIM comparison over number $N_s$ of estimations per observation (MNIST, $M = 200, N_t = 20000$), c)-f) nMSE and SSIM performance of models trained on compressed data (solid curves) as well as models trained on ground-truth data (dashed curves) $\mathcal{W}$ and $\mathcal{X}$ over c) and d) $M$ ($N_t = 20000$, MNIST), and e) and f) $N_t$ ($M = 160$, MNIST), g) and h) performance comparison on FashionMNIST.

Each block corresponds to one color channel. We use this dictionary for all estimators on celebA. The batch size to set to 32. Otherwise, all hyperparameters are chosen in the same way as for MNIST.

Generally, we did no detailed network architecture search for the proposed CSVAE since we observed no considerable change in performance by testing out different architectures.

## L Additional Results

**Robustness comparison.** The proposed CSVAE relies on approximating the outer expectation in (9) by Monte-Carlo sampling. Similar to this approximation, the baseline CSGAN also applies several random restarts, i.e., it estimates the ground-truth sample several times for a single observation and chooses the best-performing estimation by comparing their tractable measurement errors [30, 4]. In this way, both methods can be compared in terms of the number $N_s$ of repeated estimations they perform for a single observation. In Fig. 6 a) and b), we compare the nMSE and the SSIM for both with respect to $N_s$. We evaluate their performance on the MNIST dataset with $M = 200$ and $N_t = 20000$. It can be seen that the proposed CSVAE achieves already good performance for a single Monte-Carlo sample, i.e., $N_s = 1$, while CSGAN is significantly worse when only using one restart (i.e., $N_s = 1$) compared to having many restarts.

**MNIST with training on ground-truth data.** In Fig. 6 c)-f), the results from Fig. 2 a)-d) are extended with the performance of the corresponding models trained on ground-truth data, which are represented by the dashed curves. It should be noted that ground-truth information here refers to the MNIST images $\boldsymbol{x}_i$ themself. In consequence, CSGAN as well as the proposed CSGMM are trained on $\mathcal{X}$, while CSVAE is trained on $\mathcal{W}$ (cf. Section 2.1 and Appendix H). For the sake of readability, we leave out the error bars in these plots. Generally, the CSGAN, as well as the proposed CSVAE and CSGMM, benefit from the additional information during training in terms of the distortion metrics nMSE and SSIM. For the nMSE, the overall performance comparison remains the same, while for SSIM CSGAN trained on $\mathcal{X}$ outperforms CSVAE and CSGMM trained on $\mathcal{W}$ and $\mathcal{X}$, respectively. In Fig. 7, additional exemplary reconstructed MNIST images are shown for all models trained on compressed as well as ground-truth data $\mathcal{X}$ in case of the CSGAN and CSGMM and $\mathcal{W}$ in case of the CSVAE. It can be seen that perceptually, the CSGAN significantly benefits from the ground-truth

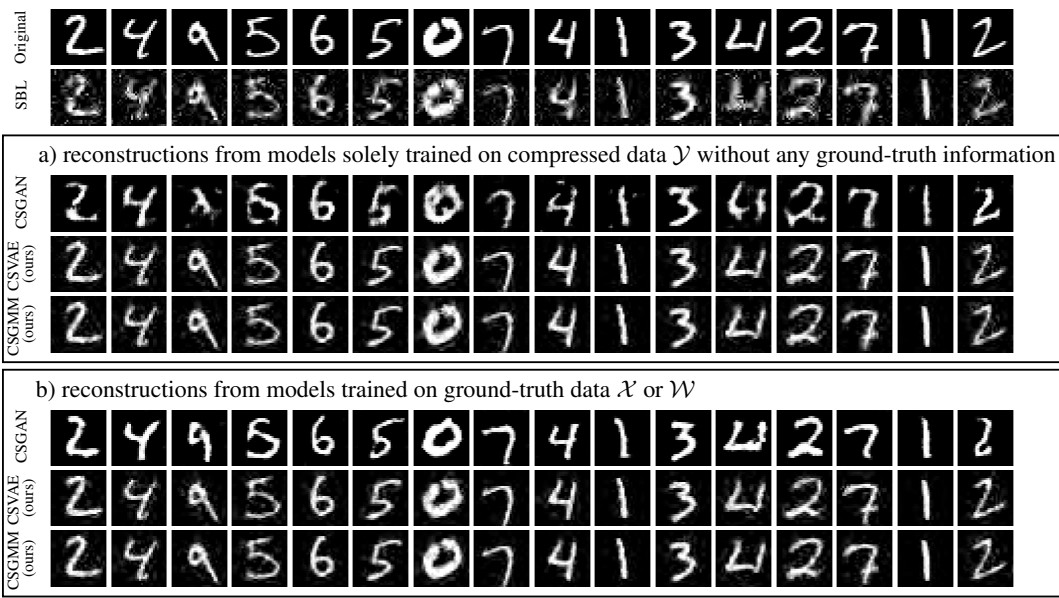

Figure 7: Exemplary reconstructed MNIST images for $M = 200$, $N_t = 20000$ from a) models, which are solely trained on compressed data (with observations of dimension $M$), and b) models, which are trained on ground truth data.

information during training, while the proposed CSVAE and CSGMM perform similarly in perception for a training set of compressed or ground-truth samples. This highlights their effective regularization effect explained in Section 3.4.

**FashionMNIST.** In Fig. 6 g) and h), the performance of SBL, CSGAN, and the proposed CSVAE and CSGMM for varying observation dimensions $M$ and fixed number $N_t = 20000$ of training samples is shown. While for small $M$, CSGAN outperforms the proposed CSVAE and CSGMM, its performance saturates for increasing $M$, and it performs worse than CSVAE and CSGMM. In this case, CSGAN's regularization to enforce the reconstruction to be in the generator's domain is beneficial for strongly compressed observations (i.e., for small $M$). In Fig. 8, exemplary reconstructed FashionMNIST images are shown.

**Additional exemplary reconstructions.** In Fig. 9, 10 and 11, additional exemplary reconstructions for the piecewise smooth functions, MNIST as well as celebA are shown.

## M  Overview of Compute Resources

All models have been simulated on an *NVIDIA A40 GPU* except for the proposed CSGMM, whose experiments have been conducted on an *Intel(R) Xeon(R) Gold 6134 CPU @ 3.20GHz*. We report the number of learnable parameters, the time used for training as well as the average reconstruction time after training for simulations with piecewise smooth functions, MNIST and celebA in Table 1, 2 and 3, respectively. The average reconstruction time of estimating $x^*$ from a newly observed $y$ has been measured for the MAP-based estimators in Appendix G. While the reported numbers give an overview of the comparison between the different tested models, it is important to note that we did not aim to fully optimize our simulations for computational efficiency.

## N  Pseudo-Code for the Training and Inference of the CSVAE and CSGMM

Algorithm 1 summarizes one iteration of the training procedure for the CSVAE. Algorithm 4 does the same for the CSGMM. In algorithm 2 and 3, the pseudo-code of the CME approximation and the MAP-based estimator using the CSVAE is presented, respectively (cf. (42) and (43)). In algorithm 5 and 6, the same is given for the CSGMM (cf. (44) and (45))

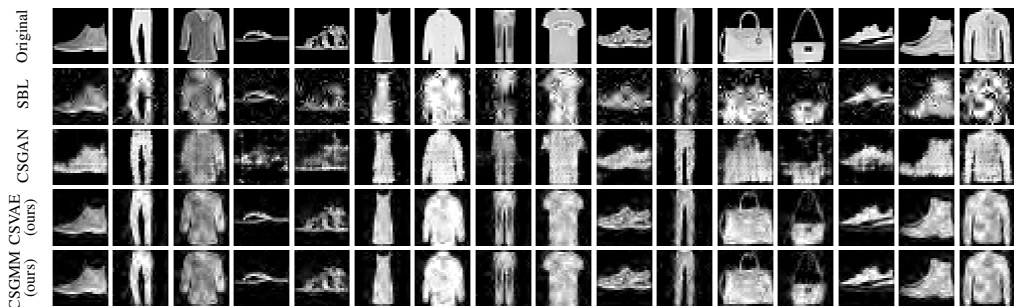

Figure 8: Exemplary reconstructed FashionMNIST images ($M = 200$, $N_t = 20000$, Fig. 6) g), h))

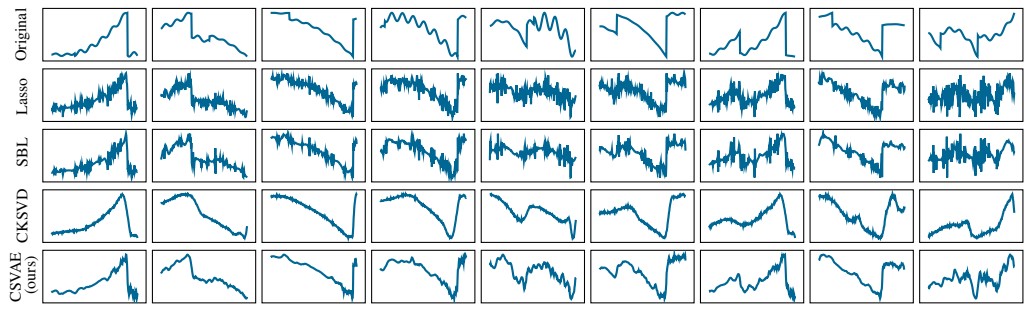

Figure 9: Exemplary reconstructed piece-wise smooth functions ($M = 140$, $N_t = 10000$, Fig. 2 f))

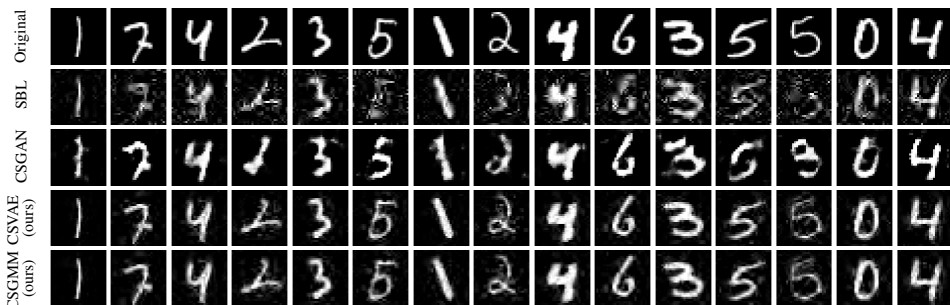

Figure 10: Exemplary reconstructed MNIST images ($M = 160$, $N_t = 20000$, Fig. 2 a))

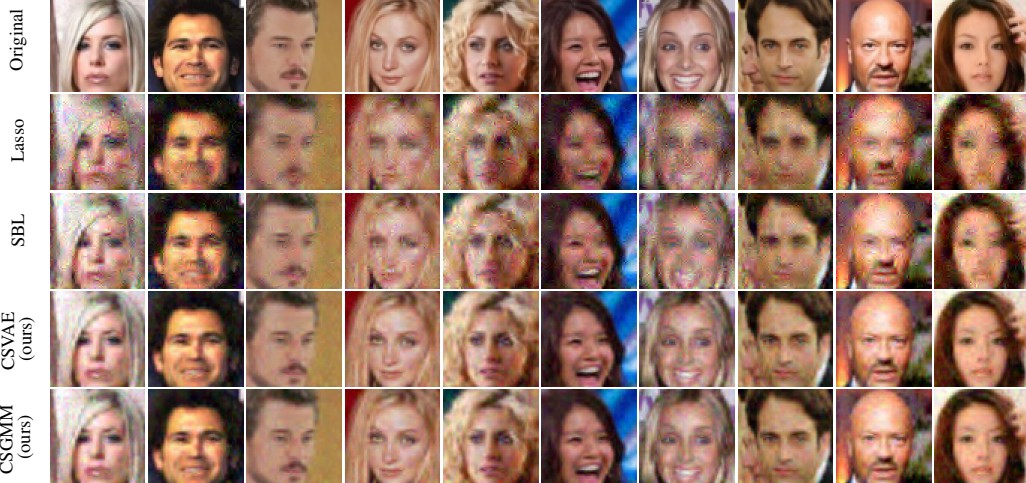

Figure 11: Exemplary reconstructed celebA images ($M = 2700$, $N_t = 5000$, Fig. 3 a))

Table 1: Resources for simulations on piecewise smooth fct. ($M = 120$, $N_t = 20000$, Fig. 2 f)).

| Model | # Parameters | Training time (hours) | Avg reconstruction time in ms |
|-------|-------------|----------------------|------------------------------|
| SBL | - | - | 137.23 |
| CKSVD | - | 0.43 | 2.38 |
| CSVAE (ours) | 109376 | 0.24 | 1.47 |

Table 2: Resources for simulations on MNIST ($M = 200$, $N_t = 20000$, Fig. 2 a)).

| Model | # Parameters | Training time (hours) | Avg reconstruction time in ms |
|-------|-------------|----------------------|------------------------------|
| SBL | - | - | 294.17 |
| CSGAN | 208817 | 24.68 | 480.81 |
| CSGMM (ours) | 46208 | 0.75 | 2.64 |
| CSVAE (ours) | 566467 | 1.48 | 2.02 |

Table 3: Resources for simulations on celebA ($M = 1800$, $N_t = 5000$, Fig. 3 a)).

| Model | # Parameters | Training time (hours) | Avg reconstruction time in ms |
|-------|-------------|----------------------|------------------------------|
| SBL | - | - | 6067.70 |
| CSGMM (ours) | 555104 | 5.33 | 329.34 |
| CSVAE (ours) | 5063138 | 17.88 | 62.59 |

---

**Algorithm 1** Update Step in the Training Phase of the CSVAE

---

**Input:** parameters in the $t$th iteration $\boldsymbol{\theta}^{(t)}$ (i.e., the decoder) and $\boldsymbol{\phi}^{(t)}$ (i.e., the encoder), batch $\mathcal{Y}_{\text{batch}}$, meas. matrix $\boldsymbol{A}$ (or corresponding batch meas. matrices $\{\boldsymbol{A}_i\}_i$), dict. $\boldsymbol{D}$, noise $\sigma^2$, optimizer $Adam_t$ in the $t$th iteration, learning rate $\lambda$

**Output:** parameters in the $(t+1)$th iteration $\boldsymbol{\theta}^{(t+1)}$, $\boldsymbol{\phi}^{(t+1)}$

**for** $i = 1$ **to** $|\mathcal{Y}_{\text{batch}}|$ **do**

    1) $\boldsymbol{\mu}_{\boldsymbol{\phi}^{(t)}}(\boldsymbol{y}_i), \boldsymbol{\sigma}_{\boldsymbol{\phi}^{(t)}}(\boldsymbol{y}_i) \xleftarrow{\text{Encoder}} \boldsymbol{y}_i$

    2) draw $\tilde{\boldsymbol{z}}_i \sim q_{\boldsymbol{\phi}^{(t)}}(\boldsymbol{z}|\boldsymbol{y}_i) = \mathcal{N}(\boldsymbol{z}; \boldsymbol{\mu}_{\boldsymbol{\phi}^{(t)}}(\boldsymbol{y}_i), \boldsymbol{\sigma}_{\boldsymbol{\phi}^{(t)}}(\boldsymbol{y}_i))$ (via reparameterization trick [37])

    3) $\boldsymbol{\gamma}_{\boldsymbol{\theta}^{(t)}}(\tilde{\boldsymbol{z}}_i) \xleftarrow{\text{Decoder}} \tilde{\boldsymbol{z}}_i$

    4) $\left( \boldsymbol{C}^{\boldsymbol{y}|\tilde{\boldsymbol{z}}_i}_{\boldsymbol{\theta}^{(t)}}(\tilde{\boldsymbol{z}}_i), \boldsymbol{\mu}^{\boldsymbol{s}|\boldsymbol{y}_i, \tilde{\boldsymbol{z}}_i}_{\boldsymbol{\theta}^{(t)}}(\tilde{\boldsymbol{z}}_i) \right) \xleftarrow{(52),(50)} \left( \boldsymbol{\gamma}_{\boldsymbol{\theta}^{(t)}}(\tilde{\boldsymbol{z}}_i), \boldsymbol{A}, \boldsymbol{D}, \sigma^2, \boldsymbol{y}_i \right)$

    5) $\mathbb{E}_{p_{\boldsymbol{\theta}^{(t)}}(\boldsymbol{s}|\tilde{\boldsymbol{z}}_i, \boldsymbol{y}_i)}[\log p(\boldsymbol{y}_i|\boldsymbol{s})] \xleftarrow{\text{Appendix D,I}} \left( \boldsymbol{\mu}^{\boldsymbol{s}|\boldsymbol{y}_i, \tilde{\boldsymbol{z}}_i}_{\boldsymbol{\theta}^{(t)}}(\tilde{\boldsymbol{z}}_i), \boldsymbol{A}, \boldsymbol{D}, \sigma^2, \boldsymbol{y}_i \right)$

    6) $\mathrm{D}_{\text{KL}}(q_{\boldsymbol{\phi}^{(t)}}(\boldsymbol{z}|\boldsymbol{y}_i)||p(\boldsymbol{z})) \xleftarrow{\text{Appendix E}} \left( \boldsymbol{\mu}_{\boldsymbol{\phi}^{(t)}}(\boldsymbol{y}_i), \boldsymbol{\sigma}_{\boldsymbol{\phi}^{(t)}}(\boldsymbol{y}_i) \right)$

    7) $\mathrm{D}_{\text{KL}}(p_{\boldsymbol{\theta}^{(t)}}(\boldsymbol{s}|\tilde{\boldsymbol{z}}_i, \boldsymbol{y}_i)||p_{\boldsymbol{\theta}^{(t)}}(\boldsymbol{s}|\tilde{\boldsymbol{z}}_i)) \xleftarrow{\text{Appendix E,I}} \left( \boldsymbol{C}^{\boldsymbol{y}|\tilde{\boldsymbol{z}}_i}_{\boldsymbol{\theta}^{(t)}}(\tilde{\boldsymbol{z}}_i), \boldsymbol{\mu}^{\boldsymbol{s}|\boldsymbol{y}_i, \tilde{\boldsymbol{z}}_i}_{\boldsymbol{\theta}^{(t)}}(\tilde{\boldsymbol{z}}_i), \boldsymbol{\gamma}_{\boldsymbol{\theta}^{(t)}}(\tilde{\boldsymbol{z}}_i) \right)$

**end for**

$L^{(\text{CSVAE})}_{(\boldsymbol{\theta}^{(t)}, \boldsymbol{\phi}^{(t)})} \xleftarrow{(15)} \{5), 6), 7)\}^{|\mathcal{Y}_{\text{batch}}|}_{i=1}$

$\left( \boldsymbol{\theta}^{(t+1)}, \boldsymbol{\phi}^{(t+1)} \right) \leftarrow Adam_t(L^{(\text{CSVAE})}_{(\boldsymbol{\theta}^{(t)}, \boldsymbol{\phi}^{(t)})}, \lambda, \boldsymbol{\theta}^{(t)}, \boldsymbol{\phi}^{(t)})$

---

**Algorithm 2** CME Approximation with the CSVAE in the Inference Phase (cf. (42))

---

**Input:** observation $\boldsymbol{y}$, encoder $(\boldsymbol{\mu}_{\boldsymbol{\phi}}(\cdot), \boldsymbol{\sigma}_{\boldsymbol{\phi}}(\cdot))$, decoder $\boldsymbol{\gamma}_{\boldsymbol{\theta}}(\cdot)$, meas. matrix $\boldsymbol{A}$, dict. $\boldsymbol{D}$, noise $\sigma^2$, cardinality $|\mathcal{Z}|$ in (42)

**Output:** CME approximation $\hat{\boldsymbol{x}}^*_{\text{CME}}$

1) $\boldsymbol{\mu}_{\boldsymbol{\phi}}(\boldsymbol{y}), \boldsymbol{\sigma}_{\boldsymbol{\phi}}(\boldsymbol{y}) \xleftarrow{\text{Encoder}} \boldsymbol{y}$

**for** $i = 1$ **to** $|\mathcal{Z}|$ **do**

    2) draw $\tilde{\boldsymbol{z}}_i \sim q_{\boldsymbol{\phi}}(\boldsymbol{z}|\boldsymbol{y}) = \mathcal{N}(\boldsymbol{z}; \boldsymbol{\mu}_{\boldsymbol{\phi}}(\boldsymbol{y}), \boldsymbol{\sigma}_{\boldsymbol{\phi}}(\boldsymbol{y}))$

    3) $\boldsymbol{\gamma}_{\boldsymbol{\theta}}(\tilde{\boldsymbol{z}}_i) \xleftarrow{\text{Decoder}} \tilde{\boldsymbol{z}}_i$

    4) $\boldsymbol{\mu}^{\boldsymbol{s}|\boldsymbol{y}, \tilde{\boldsymbol{z}}_i}_{\boldsymbol{\theta}}(\tilde{\boldsymbol{z}}_i) \xleftarrow{(50)} \left( \boldsymbol{\gamma}_{\boldsymbol{\theta}}(\tilde{\boldsymbol{z}}_i), \boldsymbol{A}, \boldsymbol{D}, \sigma^2, \boldsymbol{y} \right)$

**end for**

5) $\hat{\boldsymbol{x}}^*_{\text{CME}} = \boldsymbol{D}/|\mathcal{Z}| \sum_{i=1}^{|\mathcal{Z}|} \boldsymbol{\mu}^{\boldsymbol{s}|\boldsymbol{y}, \tilde{\boldsymbol{z}}_i}_{\boldsymbol{\theta}}(\tilde{\boldsymbol{z}}_i)$

---

---

**Algorithm 3** MAP-based Estimator with the CSVAE in the Inference Phase (cf. (43))

---

**Input:** observation $\boldsymbol{y}$, encoder $(\boldsymbol{\mu}_{\boldsymbol{\phi}}(\cdot), \boldsymbol{\sigma}_{\boldsymbol{\phi}}(\cdot))$, decoder $\boldsymbol{\gamma}_{\boldsymbol{\theta}}(\cdot)$, meas. matrix $\boldsymbol{A}$, dict. $\boldsymbol{D}$, noise $\sigma^2$
**Output:** MAP-based estimator $\hat{\boldsymbol{x}}_{\mathrm{MAP}}^*$
1) $\boldsymbol{\mu}_{\boldsymbol{\phi}}(\boldsymbol{y}) \xleftarrow{\text{Encoder}} \boldsymbol{y}$
3) $\boldsymbol{\gamma}_{\boldsymbol{\theta}}(\boldsymbol{\mu}_{\boldsymbol{\phi}}(\boldsymbol{y})) \xleftarrow{\text{Decoder}} \boldsymbol{\mu}_{\boldsymbol{\phi}}(\boldsymbol{y})$
4) $\boldsymbol{\mu}_{\boldsymbol{\theta}}^{\boldsymbol{s}|\boldsymbol{y}, \boldsymbol{\mu}_{\boldsymbol{\phi}}(\boldsymbol{y})}(\boldsymbol{\mu}_{\boldsymbol{\phi}}(\boldsymbol{y})) \xleftarrow{(50)} (\boldsymbol{\gamma}_{\boldsymbol{\theta}}(\boldsymbol{\mu}_{\boldsymbol{\phi}}(\boldsymbol{y})), \boldsymbol{A}, \boldsymbol{D}, \sigma^2, \boldsymbol{y})$
5) $\hat{\boldsymbol{x}}_{\mathrm{MAP}}^* = \boldsymbol{D}\boldsymbol{\mu}_{\boldsymbol{\theta}}^{\boldsymbol{s}|\boldsymbol{y}, \boldsymbol{\mu}_{\boldsymbol{\phi}}(\boldsymbol{y})}(\boldsymbol{\mu}_{\boldsymbol{\phi}}(\boldsymbol{y}))$

---

---

**Algorithm 4** One EM Step in the Training Phase of the CSGMM with one fixed $\boldsymbol{A}$

---

**Input:** parameters in the $t$th iteration $\{\boldsymbol{\gamma}_k^{(t)}, \rho_k^{(t)}\}_{k=1}^K$, training set $\mathcal{Y}$, meas. matrix $\boldsymbol{A}$, dict. $\boldsymbol{D}$, noise $\sigma^2$
**Output:** parameters in the $(t+1)$th iteration $\{\boldsymbol{\gamma}_k^{(t+1)}, \rho_k^{(t+1)}\}_{k=1}^K$
**for** $k = 1$ **to** $K$ **do**
   1) $\boldsymbol{C}_{k,t}^{\boldsymbol{y}|k} \xleftarrow{(17)} \left(\boldsymbol{\gamma}_k^{(t)}, \boldsymbol{D}, \boldsymbol{A}, \sigma^2\right)$
   2) $\mathrm{diag}\left(\boldsymbol{C}_{k,t}^{\boldsymbol{s}|\boldsymbol{y},k}\right) \xleftarrow{(53)} \left(\boldsymbol{\gamma}_k^{(t)}, \boldsymbol{D}, \boldsymbol{A}, \sigma^2\right)$
   **for** $i = 1$ **to** $|\mathcal{Y}|$ **do**
      3) $p_t(k|\boldsymbol{y}_i) \xleftarrow{\text{(Bayes)}} \left(\boldsymbol{C}_{k,t}^{\boldsymbol{y}|k}, \rho_k^{(t)}\right)$
      4) $\boldsymbol{\mu}_{k,t}^{\boldsymbol{s}|\boldsymbol{y}_i,k} \xleftarrow{(50)} \left(\boldsymbol{C}_{k,t}^{\boldsymbol{y}|k}, \boldsymbol{D}, \boldsymbol{A}, \sigma^2, \boldsymbol{y}_i\right)$
   **end for**
   5) $\left(\boldsymbol{\gamma}_k^{(t+1)}, \rho_k^{(t+1)}\right) \xleftarrow{\text{Lemma (3.2)}} \left(\left\{p_t(k|\boldsymbol{y}_i), \boldsymbol{\mu}_{k,t}^{\boldsymbol{s}|\boldsymbol{y}_i,k}\right\}_{i=1}^{|\mathcal{Y}|}, \mathrm{diag}\left(\boldsymbol{C}_{k,t}^{\boldsymbol{s}|\boldsymbol{y},k}\right)\right)$
**end for**

---

---

**Algorithm 5** CME Approximation with the CSGMM in the Inference Phase (cf. (44))

---

**Input:** observation $\boldsymbol{y}$, GMM $\{\rho_k, \boldsymbol{\gamma}_k\}_{k=1}^K$, meas. matrix $\boldsymbol{A}$, dict. $\boldsymbol{D}$, noise $\sigma^2$
**Output:** CME approximation $\hat{\boldsymbol{x}}_{\mathrm{CME}}^*$
**for** $k = 1$ **to** $K$ **do**
   1) $\boldsymbol{C}_k^{\boldsymbol{y}|k} \xleftarrow{(17)} \left(\boldsymbol{\gamma}_k, \boldsymbol{D}, \boldsymbol{A}, \sigma^2\right)$
   2) $p(k|\boldsymbol{y}) \xleftarrow{\text{(Bayes)}} \left(\boldsymbol{C}_k^{\boldsymbol{y}|k}, \rho_k\right)$
   3) $\boldsymbol{\mu}_k^{\boldsymbol{s}|\boldsymbol{y},k} \xleftarrow{(50)} \left(\boldsymbol{C}_k^{\boldsymbol{y}|k}, \boldsymbol{D}, \boldsymbol{A}, \sigma^2, \boldsymbol{y}\right)$
**end for**
4) $\hat{\boldsymbol{x}}_{\mathrm{CME}}^* = \boldsymbol{D}\sum_{k=1}^K p(k|\boldsymbol{y})\boldsymbol{\mu}_k^{\boldsymbol{s}|\boldsymbol{y},k}$

---

---

**Algorithm 6** MAP-based Estimator with the CSGMM in the Inference Phase (cf. (45))

---

**Input:** observation $\boldsymbol{y}$, GMM $\{\rho_k, \boldsymbol{\gamma}_k\}_{k=1}^K$, meas. matrix $\boldsymbol{A}$, dict. $\boldsymbol{D}$, noise $\sigma^2$
**Output:** MAP-based estimation $\hat{\boldsymbol{x}}_{\mathrm{MAP}}^*$
**for** $k = 1$ **to** $K$ **do**
   1) $\boldsymbol{C}_k^{\boldsymbol{y}|k} \xleftarrow{(17)} \left(\boldsymbol{\gamma}_k, \boldsymbol{D}, \boldsymbol{A}, \sigma^2\right)$
   2) $p(k|\boldsymbol{y}) \xleftarrow{\text{(Bayes)}} \left(\boldsymbol{C}_k^{\boldsymbol{y}|k}, \rho_k\right)$
**end for**
3) $\hat{k}_{\mathrm{MAP}} = \mathrm{argmax}\, p(k|\boldsymbol{y})$
4) $\boldsymbol{\mu}_{\hat{k}_{\mathrm{MAP}}}^{\boldsymbol{s}|\boldsymbol{y},\hat{k}_{\mathrm{MAP}}} \xleftarrow{(50)} \left(\boldsymbol{C}_k^{\boldsymbol{y}|\hat{k}_{\mathrm{MAP}}}, \boldsymbol{D}, \boldsymbol{A}, \boldsymbol{y}, \sigma^2\right)$
5) $\hat{\boldsymbol{x}}_{\mathrm{MAP}}^* = \boldsymbol{D}\boldsymbol{\mu}_{\hat{k}_{\mathrm{MAP}}}^{\boldsymbol{s}|\boldsymbol{y},\hat{k}_{\mathrm{MAP}}}$

---

