# OpenReview forum: "Sparse Bayesian Generative Modeling for Compressive Sensing"
_NeurIPS.cc/2024/Conference — NeurIPS 2024 poster_

### Official Review · Reviewer_34q1 · 2024-06-17

**Soundness:** 2
**Presentation:** 2
**Contribution:** 2
**Rating:** 6
**Confidence:** 3

**Summary:**

This paper introduces a new type of sparsity inducing generative prior for the inverse problem. The authors theoretically underpin our approach by proving that its training maximizes a variational lower bound of a sparsity inducing log-evidence.

**Strengths:**

This work can learn from a few corrupted data samples and, thus, requires no ground-truth information in its training phase.

**Weaknesses:**

1. The comparison methods include Lasso, CKSVD, and CSGAN, all of which are presented before 2018. The comparison methods are too old.
2. I think the author should clarify the specific application scenarios for this research direction.
3. In Figure 7, compared with CSGAN, the results of CSVAE and CSGMM are not very clear.

**Questions:**

1. In Table 2: Resources for simulations on MNIST (M = 200, Nt = 20000, Fig. 2 a)), CSGAN has much less parameters than the proposed CSVAE , why does CSGAN take much more training time?
2. Could the reconstruction results be influenced by the amount of noise?

**Limitations:**

I think the application scenes of this work is not very clear. The datasets used for comparison are very naive, colorful images with more textures should be used. And the authors should include more recent comparison methods, existing comparison methods are all presented before 2018.

---

> ### Author Rebuttal · Authors · 2024-08-06
>
> We thank reviewer 34q1 for the comprehensive review. In the following, we address the reviewer’s raised weaknesses and questions.
>
> **To Weaknesses 1:** We would very much appreciate it if the reviewer could specify which comparison methods are missing in our work. While it is difficult to track all the latest literature, we are aware of the rapidly improving diffusion-based and unfolding-based compressive sensing techniques recently published. However, to the best of our knowledge, these techniques generally rely on the assumption of being trained on many ground-truth training samples. CSGAN seems to us to be the current state-of-the-art generative prior, which also relaxes the requirements on the training data by incorporating the knowledge that the prior is to be used for compressive sensing, which is why we compare our method to CSGAN.
>
> **To Weaknesses 2:** While we do not think that we can comprehend all possible applications, we give an overview of three possible applications we can think of, in which the ability to learn from compressed data is crucial and where many state-of-the-art machine learning-based compressive sensing techniques might not be applicable.
>
> 1) ECG denoising: The sensors in wearable electrocardiography (ECG) monitoring devices generally provide noisier signals, with more artifacts compared to those typically used in the hospital [1]. However, if one wants to do patient-specific training, the ability to directly train from the data provided by these sensors comes with severe benefits as it is rather unrealistic to first capture clean ground-truth training samples in hospitals for each patient individually.
>
> 2) Electron microscopy: In electron microscopy, one generally wants to restrict the total amount of electron dose used for measuring to not interact with (change) or even destroy the measured sample of interest [2]. However, this typically leads to noisy acquisitions with low contrast and resolution. Thus, being able to learn from these corrupted acquisitions also comes with significant benefits.
>
> 3) Wireless communication: In, e.g., the current 5G wireless communication standard, mobile users during communication receive compressed observations on a frequent basis (see Eq. (1) in [3]). While current machine learning techniques in wireless communication rely on simulated data or require expensive measurement campaigns for the training data, our proposed method can directly learn from the compressed data that mobile users receive during communication.
>
> All these applications have in common that they are rather low-dimensional. Moreover, latency plays an important role in ECG denoising as well as wireless communication rendering our proposed method to be specifically interesting for those applications.
>
> **To Weaknesses 3:** These observations come from the different reconstruction aspects of CSGAN and CSVAE/CSGMM. Our estimators approximate the point conditional mean estimators, which minimizes the distortion measure MSE. Therefore, CSVAE- and CSGMM-based reconstruction emphasizes the reconstruction of details, while CSGAN emphasizes the visual contrast. This can be explained by the perception-distortion tradeoff [4]. By comparing the ground truth (first row) with the reconstructed images more closely, one can see that the CSGAN misses out on details, which are reconstructed by CSVAE and CSGMM.
>
> **To Question 1:** The different training times result from the distinct training algorithms between CSGAN and CSVAE/CSGMM. More specifically, the training of CSGAN builds on a min-max optimization problem, where additionally after each update step, a Lasso-like reconstruction algorithm has to be applied. On the other hand, the training of CSVAE optimizes the ELBO derived in Eq. (15).
>
> **To Question 2:** The noise level plays indeed an important role. In fact, since our proposed model can incorporate the noise level in the training process (see Eq. (2)), our proposed method is quite robust against additional noise. To evaluate this, we simulated different noise levels on MNIST and plotted the nMSE and SSIM results in the attached PDF (Fig. 4 a) and b)). One can see that our proposed method exhibits significantly less performance decrease for higher noise levels (i.e., smaller SNR) compared to all other baselines.
>
> **To Limitations:** The choice of low-resolution grayscale images and 1D signals in our work is mainly motivated by the exemplary applications stated above, which are all low-dimensional and can be interpreted as either 1D or grayscaled 2D signals/images. Additionally, the used datasets in our work have been considered as the benchmark datasets for our baselines.
>
> We would like to thank the reviewer again for the valuable feedback. We will revise our final paper to point out exemplary applications more specifically. Moreover, we plan to also include the simulations on varying noise levels.
>
> [1] G. Revach, T. Locher, N. Shlezinger, R. J. G. v. Sloun, R. Vullings, "HKF: Hierarchical Kalman Filtering with Online Learned Evolution Priors for Adaptive ECG Denoising," 2023, arXiv:2210.12807.
>
> [2] T.-O. Buchholz, M. Jordan, G. Pigino, and F. Jug, “Cryo-care: Content-aware image restoration for cryo-transmission electron microscopy data,” in 2019 IEEE 16th International Symposium on Biomedical Imaging (ISBI 2019), 2019, pp. 502–506.
>
> [3] W. Kim, Y. Ahn, J. Kim, and B. Shim, “Towards deep learning-aided wireless channel estimation and channel state information feedback for 6G,” Journal of Communications and Networks, vol. 25, no. 1, pp. 61–75, 2023.
>
> [4] Y. Blau and T. Michaeli, “The perception-distortion tradeoff,” in 2018 IEEE/CVF Conference on Computer Vision and Pattern Recognition. IEEE, Jun. 2018.

---

> ### Comment · Reviewer_34q1 · 2024-08-10
>
> Thank you for the careful rebuttal. I thins the authors have addressed my concerns. I hope you can revise the final paper to point out exemplary applications more specifically. I have improved my rating.

---

### Official Review · Reviewer_WoRD · 2024-07-04

**Soundness:** 3
**Presentation:** 4
**Contribution:** 3
**Rating:** 7
**Confidence:** 2

**Summary:**

The authors present an elegant new approach for dictionary based compressive sensing wherein the sparsity inducing prior is tuned from data by maximizing a lower bound on the evidence. This is a new paradigm for compressive sensing which appears to improve on reconstruction error over standard approaches and does not require solving an optimization problem at inference time.

**Strengths:**

- Approaching dictionary based compressive sensing in this way is novel as far as I'm aware. Not having to solve an optimization problem at inference time has the potential to be impactful.
- I found the paper to be very well-written. The authors did a good job of motivating their approach by relating it back to relevant background literature and discussing necessary theory.
- Their results appear to be promising and should hopefully motivate future work in this paradigm.

**Weaknesses:**

- Since their approach offers a new perspective on sparse Bayesian learning, it would have been helpful to compare the sparsity of predictions in addition to the nMSE.
- I understand that space is limited, but I found the description of the experiments to be a bit terse.
- It wasn't clear to me when a practitioner should choose CSVAE vs. CSGMM

**Questions:**

- How does your approach compare to methods for sparse Bayesian learning with proper priors; for example Louizos et. al. "Bayesian Compression for Deep Learning." NeurIPS 2017?
- Have you tried evaluating the quality of error bars from the predictive posterior so that your approach might be used in the context of UQ?
- Can you explain the reason for the numerical instabilities in the noise free simulations for some of the image datasets?
- Have you analyzed the histogram of the posterior over $s$? Can you suggest how the practitioner who is interested in a sparse $s$ might $0$ entries of the vector?

**Limitations:**

I think the authors have done a good job addressing the potential limitations of their approach in section 3.4. It would be interesting to analyze how well generative priors trained on a particular dataset transfer to similar datasets.

---

> ### Author Rebuttal · Authors · 2024-08-06
>
> We would like to thank reviewer WoRD for the positive review and the appreciation of our work. In the following, we address the reviewer’s raised weaknesses and questions.
>
> **To Weaknesses (bullet point 1):** We thank the reviewer for this suggestion and give it serious consideration for the final manuscript.
>
> **To Weaknesses (bullet point 2):** As the final manuscript is allowed to have one additional page, we plan to reorganize the experimental section and describe the simulations in more detail.
>
> **To Weaknesses (bullet point 3):** There are a few aspects to consider when choosing between CSGMM and CSVAE comprising the following:
>
> 1) After training CSGMM with training samples observed with one fixed measurement matrix, it is possible to apply the model to reconstruct signals from different measurement matrices even with different dimensions. This is not possible with CSVAEs as the encoder is trained measurement matrix specific.
>
> 2) When the training set has been observed using varying measurement matrices, the covariance matrices in Eq. (17) become sample-dependent. Moreover, as the EM algorithm requires all training samples for one update step, storing all these sample-specific covariance matrices might lead to memory-related issues for the CSGMM. This issue does not arise with CSVAEs since there, the update step is done via small training batches.
>
> 3) For the online reconstruction, CSGMM has to evaluate $p(k|\mathbf{y})$ for all components $k$. For very high dimensions, this can get computationally more expensive than the simple forward operation through the equivalent CSVAE's NN encoder.
>
> **To Question 1:** In the mentioned paper, the priors are assigned directly to the weights of the neural network. However, in our neural network-based parameterization of CSVAE, we keep the parameters (i.e., $\mathbf{\theta}$) deterministic but rather combine the sparse Bayesian learning framework with the output $\mathbf{\gamma}_{\mathbf{\theta}}(\mathbf{z})$ of the CSVAE by interpreting it as the diagonal of a zero-mean Gaussian with diagonal covariance matrix. From our perspective, both settings are rather different and are difficult to compare to each other. We hope the reviewer agrees on this point with us.
>
> **To Question 2:** While we think, that the idea to explicitly evaluate the error bars of the posteriors is interesting, we did not try this out. We appreciate the idea and will give it serious consideration for the final manuscript. Additionally, in the attached PDF, we now included error bars in the results for the nMSE and SSIM.
>
> **To Question 3:** We implemented CSVAE in pytorch. The numerical instabilities concern the backpropagation of Eq (24) in our work, which includes the pseudoinverse. Although (24) is theoretically differentiable w.r.t. $\mathbf{\theta}$, it turned out that utilizing the equivalent (8) instead of (24) with some small artificial noise variance (equivalent to 40dB SNR) led to stable training, while directly using (24) sometimes resulted in numerical issues.
>
> **To Question 4:** From our perspective, this question seems to relate to Question 2. We did not analyze the posterior's histogram. We think, there are several possibilities to approximate the compressible estimate of $\mathbf{s}$ by a sparse one. One way could be to decide on some decision threshold. We refer to [1] for more details.
>
> We would like to thank the reviewer again and give the reviewer's suggestions regarding the sparsity and posterior analysis a serious consideration for the final manuscript. Additionally, we will extend the Appendix by a discussion about when to choose CSGMM over CSVAE and vice versa.
>
> [1] G. Dziwoki, M. Kucharczyk, "On a Sparse Approximation of Compressible Signals," 2020, Circuits Syst. Signal Process. 39(4): 2232-2243

---

> ### Comment · Reviewer_WoRD · 2024-08-08
>
> Thank you for your thoughtful response. I think extending the Appendix to include
> a more concrete discussion of CGSMM versus CSVAE would be helpful.
>
> While not a requirement, I hope you do
> end up including some analysis of the posterior and sparsity since this will
> make your work relevant to a broader community.

---

### Official Review · Reviewer_y1C8 · 2024-07-15

**Soundness:** 2
**Presentation:** 2
**Contribution:** 3
**Rating:** 5
**Confidence:** 4

**Summary:**

The paper introduces a novel training algorithm for generative models used as priors in linear inverse problems, with a specific focus on compressed sensing. The authors propose a training principle that regularizes the prior to learn a sparse representation of the signal of interest, implemented in Variational Autoencoders (VAEs) and Gaussian Mixture Models (GMMs). This is achieved by learning two posterior distributions p(y∣s) and p(s∣z) over a set of parameters through optimization of the Evidence Lower Bound (ELBO) derived in Equation 15.

During the inversion phase, the true signal is estimated by sampling from the posterior distribution E[x∣y] as given in Equation 9. Unlike many contemporary generative priors, this method does not require solving an optimization algorithm to achieve the estimate. This approach simplifies the inversion process, making it more computationally efficient.

The performance of these implementations is validated on datasets containing different types of compressible signals. The results demonstrate the effectiveness of the proposed method in accurately reconstructing signals. Additionally, the paper provides theoretical support on the tightness of the lower bound of the estimate, further validating the robustness and reliability of the proposed approach.

**Strengths:**

The proposed method appears to be novel and aligns with an intriguing line of work that focuses on training generative priors with the intention of using them for inverse problems.

Section 2 of the paper is well-crafted, effectively elucidating the connection between the proposed method and existing literature.

For the given baselines, the proposed method yields better reconstructions of the true signal, as verified by commonly used metrics and visual inspection.

**Weaknesses:**

The authors provide theoretical motivation; however, they do not address how the potential estimation of the measurement y impacts the estimation of the true signal x. Additionally, building on this point, the CSVAE may experience representation error, as noted in [5], which is not mentioned as a limitation. While the proposed method does improve upon [5], this remains a source of error.


Section 4 is limited in the following ways:

* The only metric reported for natural images is the normalized mean square error, while other commonly used metrics, such as PSNR and SSIM, are not included.

* Given the computational focus of this paper, it is concerning that all experiments were conducted on low-resolution grayscale images, especially considering the A40 GPU budget.

* The baselines appear to be less contemporary compared to current literature on compressed sensing inversion algorithms. [Normalizing Flow](https://proceedings.mlr.press/v119/asim20a/asim20a.pdf) and [Diffusion Models](https://openreview.net/pdf/4f6f0e2347a3d6f9a88b39e445f77c1e7503064e.pdf) have largely replaced GANs due to issues with representation error. Similar to the proposed method, Diffusion Generative Priors do not require gradient-based approaches to estimate the posterior. There are hyperlinks to the work in the responses.

* Considering this paper is training generative before be used downstream, there seems to be no reported metrics about training other than computational resources.

**Questions:**

* Could the authors please provide perceptual metrics (i.e. SSIM, LPIPS) for the reconstruction comparison experiments in Figures 2 and 3?
* Could the authors elaborate on the computational overhead provided in footnote 4 and explain the reason for utilizing only grayscale images when reporting having a computational budget of NVIDIA A40 GPU for CSVAE?
* Could the authors please explain the advantages the proposed method has over contemporary generative prior such as normalizing flow and diffusion-based priors?
* One concept I am unclear about is how do you check if the proposed method is overly biased towards the measurements y? This issue might arise because the conditional distribution P(y|s)  has equally likely probabilities when Y=AS, but does not guarantee S will be consistent with training samples depending on how sparse s is in practice.

**Limitations:**

Please refer to the weakness portion of the review.

---

> ### Author Rebuttal · Authors · 2024-08-06
>
> We thank reviewer y1C8 for the detailed review. In the following, we address the reviewer’s raised weaknesses and questions.
>
> **To Summary:**
> We would like to point out that we do not learn $p(\mathbf{y}|\mathbf{s})$. In fact, keeping $p(\mathbf{y}|\mathbf{s})$ fixed by a pre-known dictionary is a key property of our proposed method. Instead, we learn the parameters of a statistical model for $p_{\mathbf{\theta},\mathbf{\delta}}(\mathbf{s})$. Moreover, we do not reconstruct $\mathbf{x}^*$ by posterior sampling, but rather approximate the point conditional mean estimator. This estimation methodology is different from the diffusion-based posterior sampling methods or the MAP-like GAN-based reconstruction which the reviewer might be familiar with.
>
> **To Weaknesses (before bullet points):** The representation error of GAN-based compressive sensing originates from the discrepancy between the learned GAN’s range (i.e., its learned manifold) and the ground-truth signal, which is supposed to be reconstructed. Our proposed method learns no manifold but rather a parameterized statistical model, which is why our proposed method does not experience this source of error.
>
> **To Weaknesses (bullet point 1 and Question 1):** The PSNR is a function of the nMSE, which is why the PSNR encodes the same information as the nMSE, and including both to us seems to provide no additional information. However, we agree with the reviewer that the SSIM provides additional information, which is why we plotted the SSIM for Fig. 2 and 3 in the attached pdf (Fig. 3). Our proposed methods either perform comparably or outperform CSGAN in all settings and outperform all other baselines.
>
> **To Weaknesses (bullet point 2) and Question 2:** The choice of low-resolution grayscale images and 1D signals in our work is motivated by the exemplary applications, we have in mind for our proposed method. We think that there are several applications that crucially depend on the ability to learn from strongly compressed signals and exhibit signals/images in this dimensional regime. For example, in wireless communication (either 1D or 2D low-dimensional) as well as in ECG denoising (1D signals) in wearable technology, data is typically low-dimensional, noisy, and compressed, and ground-truth training data is difficult to obtain. Moreover, in those applications, latency plays a crucial role, which is why we think that our proposed method matches those applications. However, we decided on images and generic 1D signals in our work, as these are the benchmark datasets used for our baselines, and we wanted to introduce our method independent from specific applications. We refer to our answer to reviewer 34q1 (Reviewer 4) - Weaknesses 2) for more details about exemplary applications.
> Having said that, there is indeed a memory-related limitation for our proposed method related to the computation and processing of the calculated covariance matrices. Sparse Bayesian learning-based techniques generally share this limitation. While it is possible to circumvent the explicit computation of the posterior covariance matrices in Eq. (8) for training the CSVAE and CSGMM, the covariance matrices in Eq. (17) have to be computed and inverted. This is also, what we refer to in footnote 4. In the final version of the paper, we will revise our limitations and explicitly discuss this point in more detail as this property might limit the application of our model in extremely high-dimensional image settings. To overcome this issue and combine it with efficient Sparse Bayesian learning methods is considered future work.
>
> **To Weaknesses (bullet point 3) and Question 3:** To our best knowledge, contemporary generative priors for compressive sensing generally rely on the assumption of being trained on many ground-truth training samples. However, as pointed out in our previous paragraph, there are several applications in which this assumption is difficult or even not possible to realize. In contrast, our proposed method does not require ground-truth training samples in the training phase and, thus, can be applied for those applications. CSGAN
> seems to us to be the current state-of-the-art generative prior, which also relaxes the requirements on the training data by incorporating the knowledge that the prior is to be used for compressive sensing. This is why we compare our method with CSGAN.
>
> **To Weaknesses (bullet point 4):** In the attached PDF (Fig. 4 c) and d)), we included the tracking of the objective functions for CSVAE and CSGMM as an additional metric for successful training.
>
> **To Question 4:** Unfortunately, we cannot fully comprehend the reviewer’s question. Can the reviewer specify what is meant by ”equally likely probabilities when $\mathbf{y} = \mathbf{A}\mathbf{s}$”? The reviewer might refer to a distributional shift between the training and test samples. While we think that it is an important question whether the proposed method is robust against distributional shifts in the test set, we consider this question to be out of the scope of our work as this paper is supposed to introduce the general concept and demonstrate good performance on benchmark datasets.
>
> We would like to thank the reviewer again. We will revise our final paper and plan to explicitly address the potential limitations with very high-dimensional data, which Sparse Bayesian learning generally suffers from. Moreover, we plan to also include the results on the SSIM distortion metric.

---

> ### Comment · Reviewer_y1C8 · 2024-08-12
>
> I would like to thank the reviewers for their detailed response, I will update my score accordingly based on the rebuttal.

---

> ### Comment · Reviewer_y1C8 · 2024-08-12
> **Further Questions**
>
> Yes, I agree with both points "to our best knowledge, contemporary generative priors for compressive sensing generally rely on the assumption of being trained on many ground-truth training samples" and "the ability to learn from strongly compressed signals and exhibit signals/images in this dimensional regime." However, one of the datasets used is MNIST, which has sufficient data for diffusion-based prior or normalizing flows and these priors are not limited to images they can perform inference on 1-D signal. Please correct me if something was done differently to the training set that could not be adopted for this framework.  Furthermore, even in the case where there is limited data, there are untrained priors such as [Deep Decorder](https://arxiv.org/pdf/1810.03982) or [Deep Image Prior](https://arxiv.org/pdf/1711.10925) that can be used as comparisons.
>
>
> Based on the response above you have implicity answered my question about the equally likely probabilities along the line of Y=AS.

---

> ### Author Response · Authors · 2024-08-13
> **Answer to Reviewer y1C8**
>
> We would like to thank the reviewer for commenting on our rebuttal. We agree with the reviewer that the MNIST dataset generally provides enough ground-truth training samples for training state-of-the-art machine learning-based compressive sensing techniques. However, we indeed modified the dataset for training our proposed method. In all our simulations in the main paper (Fig. 2 and 3) and most results in the Appendix (all except the dashed results in Fig. 6), the parameter $M$ does not only refer to the dimension of the compressed observation, which is to be reconstructed but also to all training samples (Section 4.1 - Measurement matrix and evaluation metric). For example, all training samples in Fig 2 a) have been compressed by a Gaussian measurement matrix to have dimension $M$ before being used for training, which is why none of the state-of-the-art methods trained on ground truth data could have been used in all these settings. We will revise our work to make this more prominent in our final version.
>
> We agree that Deep Image Prior and Deep Decoder could have been considered as further baselines. Both are cited in our introduction. However, we decided against both as these methods only apply to natural images and cannot be trained [1], whereas our setting is about reconstructing any compressible type of signals based on learning from corrupted training samples. Therefore, CSGAN and CKSVD seem to us to be the closest baselines from the "generative model"- and "dictionary learning"-community. We hope the reviewer agrees with us that both untrained neural-network baselines are, thus, not required for our work.
>
> [1]  D. Ulyanov, A. Vedaldi, and V. Lempitsky, “Deep image prior,” International Journal of Computer Vision, vol. 128, no. 7, p. 1867–1888, Mar. 2020.

---

### Official Review · Reviewer_K63t · 2024-07-26

**Soundness:** 3
**Presentation:** 2
**Contribution:** 2
**Rating:** 6
**Confidence:** 5

**Summary:**

The paper proposes a set of methods to learn a generative model over the sparse representations of  a signal. The dictionary basis is fixed and given, and the signal of interest is assumed to be sparse in this basis. Therefore, the generative model learns to provide such sparse representations. The method learns the generative prior from the noisy and compressed observations. The authors leverage the notion of conditional Gaussianity and propose VAE and GMM based models for learning such priors. The choice enables the inference with only forward operation.

**Strengths:**

The idea of using purely forward operations for inference is quite nice, since most of the solutions based on generative priors require gradient descent on the latent space and suffer from latency among other problems. Furthermore, learning the prior from compressed observations makes the approach more practical as it does not need ground truth signal.

**Weaknesses:**

Overall, the paper combines some of the existing ideas from prior works (for example posterior updates from SBL, VAE ELBO derivations), and therefore, the novelty seems marginal. To be more concrete, it *seems* straightforward that if one wants to do variational Bayes, then the VAE style decomposition is the standard approach. If there are any additional challenges, the authors should clarify.

The idea of forward pass only inference has been tried before (see for example the paper “Solving Linear Inverse Problems Provably via Posterior Sampling with Latent Diffusion Models” NeurIPS 2023).

Learning from the compressed samples can be compelling although not new as mentioned by the authors themselves (for example CSGAN). The proposed method requires further elaboration on the loss with respect to learning from uncompressed measurements substantiated with more experiments.

The experiment section is a bit thin given the limited technical novelty, particularly with respect to the baselines. Given a lot of prior works in the literature on learning based solvers for inverse problems (for example, all variations of generative priors based on flows and diffusions, unfolded algorithms like lista and its variations), the authors have a more difficult task of placing their contribution within the prior work. The paper is not very convincing in showing what is the unique direction in which the field is moved forward.

**Questions:**

Some of my questions about the novelty was implicit in my comments above.
-	Can authors provide some comment on the performance loss with respect to uncompressed generative prior?
-	How many MC samples are used at inference for (9)?
-	Generative priors generally suffer from convergence issues and require multiple restarts. Could the authors comment if their forward-only operation has a similar issue?
-	It would be good to incorporate error bars in the plots and numerical results. Given the MC step, I expect a higher variance for the proposed approach.

**Limitations:**

See comments above.

---

> ### Author Rebuttal · Authors · 2024-08-06
>
> We thank reviewer K63t for the thorough review and the appreciation of the proposed method’s ability to reconstruct via a single forward operation. In the following, we address the raised weaknesses and questions.
>
> **To Weaknesses (first paragraph):**
> We agree with the reviewer that the idea of realizing variational Bayes via a VAE-style decomposition is straightforward. However, this is not our only technical contribution and there are additional challenges and contributions from our side.
>
> 1) One technical contribution from our side is to extend the statistical model of SBL for compressive sensing in Eq. (2) by an additional (conditional) latent variable $\mathbf{z}$ with arbitrarily parameterized distribution (Eq. (7)). While this extension might look minor, this idea enables the learning of a nontrivial sparsity inducing prior with training data. In contrast, the original SBL for compressive sensing in Section 2.2 cannot learn from data but rather applies the EM algorithm to adjust a naive prior during inference. To our best knowledge, we propose the first model, which incorporates SBL for compressive sensing while simultaneously being able to learn from training data in an initial offline phase.
>
> 2) We also rigorously prove that our introduced statistical model in Eq. (7) is sparsity inducing (Theorem 3.1). We consider this as
> a further technical contribution.
>
> 3) The idea of using a VAE-style decomposition might be straightforward. However, we think that it is not trivial to show that there exists a tractable ELBO for the statistical model in Eq. (7) (which is different from the standard VAE ELBO). We derive this ELBO from (12) to (15) with additional closed-form formulas in Appendix D. Moreover, for the CSGMM, we derive the closed form of its m-step in Lemma 3.2, which solves the non-trivial optimization problem in Eq. (30).
>
> Altogether, we would like to kindly disagree that our work has ”limited technical novelty” and hope the reviewer agrees on this point.
>
> **To Weaknesses (fourth paragraph):**
> We understand the reviewer’s perspective that our baselines might seem to ignore some of the state-of-the-art compressive sensing methods. We are aware of the rapidly improving diffusion-based and unfolding-based compressive sensing techniques recently published. We also partially cite the corresponding pioneering work (such as the ALISTA paper) in our introduction. However, these state-of-the-art models have to learn from lots of ground-truth data. We want to emphasize that we neither aim nor claim to outperform these techniques as we think that it is illusory to expect a model, which is trained on a few and heavily compressed data samples (as our model), might outperform state-of-the-art pre-trained techniques trained on many ground-truth samples. From a (natural) image processing perspective, the capability to learn from compressed data might seem minor as many ground-truth natural images are often accessible. However, we think there are several applications that strongly rely on this ability and being able to outperform what seems to us as the state-of-the-art model in this regime (CSGAN) is certainly a contribution from our perspective. We additionally would like to refer to our answer to 34q1 (Reviewer 4) - Weaknesses 2) for exemplary applications where the currently available state-of-the-art models are not applicable, as this reviewer asked explicitly for applications. Having said that, we will revise our work in the final version to better place our work in the current research area.
>
> **To Question 1 and Weaknesses (third paragraph):**
> In Appendix I, we included the comparison between CSGAN and the proposed CSGMM and CSVAE trained on ground-truth and compressed data. Moreover, in the original work of CSGAN, it was shown that CSGAN can outperform the standard GAN when being trained on ground-truth data for compressive sensing. Since our main focus lies on the capability to train from compressed data, the informational gain from further baselines trained on ground-truth data seems to us rather limited (see also our previous answer). We hope the reviewer agrees with us on this.
>
> **To Question 2 and Question 4:**
> The number of Monte Carlo samples is 64 (Section 4.1 - Hyperparameters). While we understand the reviewer’s intuition of the CSVAE estimator to exhibit high variance, the opposite is the case. In fact, the proposed approach is highly robust w.r.t. the Monte Carlo approximation and already one/a few sample(s) are sufficient for approximation. In fact, CSGAN also relies on (Monte-Carlo-like) estimating $\mathbf{x}^*$ several times, which is why it is possible to explicitly compare both methods in this regard. In Fig. 1 (left) of the attached pdf, the nMSE over MC samples for CSVAE and CSGAN is shown for MNIST reconstructions with observations of dimension 200. It can be seen that taking more than one sample is almost negligible for CSVAE, while only one trial for CSGAN performs significantly worse than computing many trials. We additionally agree with the reviewer that error bars help the reader evaluate the results, which is why we included error bars representing the standard deviations of the estimations. In Fig. 1 (right) and Fig. 2 of the attached PDF, we included the same error bars in all the plots of our main paper.
>
> **To Question 3:**
> As shown in Fig. 1 (left) in the attached PDF, the inference of our proposed method is robust and requires no multiple starts. This property is based on the differences between the estimation methodology of GANs and our proposed methods. While GANs enforce their estimation to lie on a learned manifold, our proposed method regularizes the estimation by means of a probabilistic prior and aims to approximate the conditional mean estimator.
>
> We thank the reviewer again. Based on the feedback, we will revise our paper to place our work better in the research area as well as include error bars and the simulations regarding the MC samples.

---

> > ### Comment · Reviewer_K63t · 2024-08-12
> > **Follow-up on the answers**
> >
> > I would like to thank the authors for taking time to answer my questions and comments. I will discuss some of the responses a bit further, and that, with the intention of building more confidence and increasing my score, so hopefully the authors can bear with the process, and their time is well spent at the end.
> >
> > * To begin, I agree with the authors that “the capability to learn from compressed data” is not minor, as I mentioned in my original review.
> > * I am also satisfied with the answers to my questions regarding MC samples, error bars, and convergence issues.
> > * **Regarding deriving ELBO from SBL:** As far as I can see, the inequality in (7) is the standard step in ELBO bounds. The decomposition in (13) follows from the basic probability decomposition for $\mathbf{z}, \mathbf{s}$  and $\mathbf{y}$ (given in Appendix B). Then, the computation of the bound follows from MC sampling like classical VAE (also referenced by the authors themselves, namely ref. 36), and the tractable computation of some terms in Appendix D. The derivations of Appendix D (eq. 26-29) are not exactly the same as VAE paper but builds very similarly on Gaussian likelihood assumption and the linearity of expectation.  So, unfortunately, I still feel that these derivations are based on standard linear algebra and probability techniques, very similar to the standard VAE. We can therefore focus on other comments regarding the novelty.
> > * **Regarding “1. One technical contribution from our side is to extend the statistical model…”:**
> > I agree with the reviewer that the proposed method enables “the learning of a nontrivial sparsity inducing prior with training data”, however the actual contribution is “the learning of a nontrivial sparsity inducing prior” from **compressed measurements**, as the authors emphasize (although it is not so clear from the title and not so pronounced in the abstract).
> >
> > So, the question is: does the method perform equally better from uncompressed data? If not, what makes the method better in case of learning from compressed data? I feel combining two contributions (namely learning a nontrivial sparsity prior AND learning from compressed data) and selecting the baselines accordingly makes the message convoluted: is the paper about a better generative prior or about learning from compressed data? Answering such questions is important for the paper, which is proposing a general method for solving a particular problem.

---

> ### Author Response · Authors · 2024-08-13
> **Answer to Reviewever K63t**
>
> We thank the reviewer for the reviewer's comments on our rebuttal and for the willingness to increase the score if all of the reviewer's concerns are addressed.
>
> Moreover, we thank the reviewer for pointing out that our contribution might be misunderstood as we focus more on the learning aspect in our abstract and contributions instead of the "learning from compressed measurements"-aspect.
> It seems to us that the key aspects of our contributions depend on the individual background of the reader.
> Coming from a "SBL for compressive sensing" perspective, emphasizing the ability to learn from data in an initial training phase might be more natural. However, we agree that for the "generative models for compressive sensing" community, the additional aspect of being able to learn from compressed measurements is crucial to accurately putting our work's contribution into concurrent literature.
>
> In Fig. 6 and 7, we compare CSGAN, CSVAE, and CSGMM trained on uncompressed (ground-truth) data (dashed curves) as well as compressed data (solid curves). Perceptually, CSGAN benefits more from uncompressed (ground-truth) information during training than our proposed CSVAE and CSGMM. In the case of the distortion metric nMSE, all approaches perform approximately equally better when being trained on uncompressed data. In total, the performance gains to the CSGAN baseline seem to be more prominent when it comes to being trained on compressed training samples.
>
> From our perspective, this indicates that our proposed approach is particularly relevant in those cases where only compressed training samples are accessible and/or the online (inference) latency plays a crucial role. The intuition behind why the approach is capable of successfully learning from compressed data is the following:
>
> After each parameter update step during training, the proposed CSGMM and CSVAE do the following:
>
> - They estimate the corresponding ground-truth samples in the training data (Eq. (8) right side)
> - Additionally and arguably even more importantly, they quantify the uncertainty of this estimation in terms of the estimations' covariance matrix (Eq. (8) left side).
> - After doing so, both proposed models incorporate both (the estimation as well as the error quantification) in the subsequent update step. In the case of CSGMM, this is done by its corresponding M-step (Lemma 3.2). In the case of CSVAEs, this is done in its modified reconstruction loss in Appendix D, as well as the additional KL divergence in Eq. (15), which does not appear in standard VAEs.
>
> This mechanism (especially the aspect of quantifying the estimation's error and incorporating it in the next update step) does not appear in CSGAN's methodology of learning from compressed data. This is because CSGAN estimates the ground-truth samples after each update step but only provides these estimations to its discriminator without considering any uncertainty. From an intuitive point of view, this is the reason why our proposed approach seems to be superior when it comes to compressed training samples.
>
> We would like to thank the reviewer again, and based on the reviewer's comments, we will revise our work to better phrase our work's contribution regarding learning from compressed measurements. Moreover, we will consider extending our discussion section to explicitly discuss the intuitive learning aspects of our proposed method described above.

---

> > ### Comment · Reviewer_K63t · 2024-08-14
> > **Thanks for the response**
> >
> > I would like to thank the authors for the answer. I increased my score to weak accept.
> >
> > I feel the above discussion can be helpful to include in the revised version of the paper, particularly to make the story more coherent. They might even think about reflecting it in the title and abstract if permitted. I still feel that the story of the paper is about learning from compressed measurements, as it does not have a detailed case study on uncompressed measurements. Their arguments that SBL is particularly useful because of incorporating uncertainties and because of estimating the ground truth samples is compelling to me. As a stretch goal, the authors could think of ablation experiments to verify this hypothesis (for example by freezing the variance params in SBL  - not sure if it makes sense though).

---

### Author Rebuttal · Authors · 2024-08-06

Dear Program Chair, Senior Area Chair, Area Chair and Reviewers,

We would like to thank you for taking the time to review our paper and for the valuable feedback. For our response, please refer to our point-by-point responses to each reviewer below.

You will also find attached a PDF with additional simulation results, in which the plots for the corresponding reviewers are marked with blue boxes and their names.

Sincerely,

The authors.

---

### Comment · Area_Chair_SWUM · 2024-08-09
**Please review the rebuttal and respond**

Dear Reviewers,

Thank you once again for your time and effort in reviewing for NeurIPS 2024.

The authors have already submitted their rebuttal to your comments, and the discussion phase has now begun. For those who have not yet responded, please review the authors' responses carefully at your earliest convenience. Your timely feedback on whether your previous concerns have been adequately addressed is crucial to the reviewing process.

Thank you very much!

Best regards,

Area Chair

---

### Decision · Program_Chairs · 2024-09-25

**Decision:**

Accept (poster)

**Comment:**

This paper proposed one compressed sensing method with sparsity inducing conditionally Gaussian generative models. All the reviewers agree that this idea is novel and interesting. After my own reading, I also agree with that and thus I recommend acceptance. Nevertheless, as pointed out by most reviewers, especially Reviewer 34q1 and WoRD,  the authors should carefully revise the manuscript as required/suggested in the rebuttal and discussions to improve the quality, which is very important.